**EMBO** *reports*

# Structural insights into human zinc transporter ZnT1 mediated Zn²⁺ efflux

Yonghui Long [ID] [1,2,6], Zhini Zhu [ID] [1,2,6], Zixuan Zhou[1,2,6], Chuanhui Yang [ID] [1,2,6], Yulin Chao[1,2,6], Yuwei Wang [ID] [1,2], Qingtong Zhou[3], Ming-Wei Wang [ID] [4,5] & Qianhui Qu [ID] [1,2 ✉]

## Abstract

Zinc transporter 1 (ZnT1), the principal carrier of cytosolic zinc to the extracellular milieu, is important for cellular zinc homeostasis and resistance to zinc toxicity. Despite recent advancements in the structural characterization of various zinc transporters, the mechanism by which ZnTs-mediated Zn²⁺ translocation is coupled with H⁺ or Ca²⁺ remains unclear. To visualize the transport dynamics, we determined the cryo-electron microscopy (cryo-EM) structures of human ZnT1 at different functional states. ZnT1 dimerizes via extensive interactions between the cytosolic (CTD), the transmembrane (TMD), and the unique cysteine-rich extracellular (ECD) domains. At pH 7.5, both protomers adopt an outward-facing (OF) conformation, with Zn²⁺ ions coordinated at the TMD binding site by distinct compositions. At pH 6.0, ZnT1 complexed with Zn²⁺ exhibits various conformations [OF/OF, OF/IF (inward-facing), and IF/IF]. These conformational snapshots, together with biochemical investigation and molecular dynamic simulations, shed light on the mechanism underlying the proton-dependence of ZnT1 transport.

**Keywords** HZinc homeostasis; Zinc Transporter; ZnT1/SLC30A1; ZnT3/SLC30A3; H⁺/Zn²⁺ Exchange
**Subject Categories** Membranes & Trafficking; Structural Biology

## Introduction

The ubiquitously distributed zinc ions (Zn²⁺) play fundamental roles in many physiological functions (Fukada et al, 2011). Binding to ~10% proteins encoded by the human genome, Zn²⁺ can act as a structural, catalytic and/or regulatory cofactor, as well as a signaling messenger (Yamasaki et al, 2007;). Nutritional zinc is critical for normal growth and development, nerve function, immune response, and oral health (Maret, 2013; Uwitonze et al, 2020). Both zinc deficiency and excessive zinc toxicity would lead to the

disruption of normal cellular functions, and consequently to a wide variety of human health problems, including diabetes, cancers and Alzheimer's disease (Fukada et al, 2011; Chasapis et al, 2012; Prasad, 2014). The cellular Zn²⁺ homeostasis therefore needs to be precisely regulated. Two groups of transmembrane solute carriers, SLC30/ZnT and SLC39/ZIP, are deployed to mobilize Zn²⁺ across biogenic membranes, with 10 ZnTs and 14 ZIPs functionally characterized in human cells (Kambe et al, 2015). ZIPs increase the cytosolic Zn²⁺ by either importing from the extracellular space or releasing the stored Zn²⁺ from subcellular compartments, whereas ZnTs remove Zn²⁺ out of the cytoplasm and maintain physiologically low free Zn²⁺. The cooperation between ZnTs and ZIPs thus enables the cells to regulate processes responsive to hundreds of picomolar-free cytosolic Zn²⁺ concentrations (Maret and Li, 2009).

SLC30A1/ZnT1 is the founding member of SLC30/ZnT family (Palmiter and Findley, 1995), which belongs to the cation diffusion facilitator (CDF) superfamily that transport divalent transition metal cations in all three kingdoms of life (Montanini et al, 2007). Unlike most ZnTs that reside at the membranes of subcellular compartments, ZnT1 predominately localizes to the plasma membrane and is the major Zn²⁺ extruder to protect cells from zinc toxicity (Palmiter, 2004; Kambe, 2011). Mice with homozygous Znt1 gene knockout died in utero soon after implantation, and heterozygous Znt1⁺/⁻ female mice were prone to abnormal development, suggesting an unreplaceable role of ZnT1 during early embryonic development (Andrews et al, 2004). Altered expression levels of ZnT1 have been linked to Alzheimer's disease and cancers (Lovell et al, 2005; Lazarczyk et al, 2008; Lyubartseva et al, 2010; Lehvy et al, 2019; Yang et al, 2023). Activation of Ras-Raf-ERK signaling pathway can be facilitated by ZnT1 interaction, which might confer cardioprotective effect after ischemia-reperfusion (Bruinsma et al, 2002; Jirakulaporn and Muslin, 2004). In addition, ZnT1 has been implicated in inhibition of the L type calcium channel (Beharier et al, 2007; Levy et al, 2009; Shusterman et al, 2017) and in regulating endogenous zinc inhibition of NMDA receptor signaling activity (Krall et al, 2020, 2022). Recent studies also showed that ZnT1 can localize to intracellular compartments (Abdo et al, 2021), such as mitochondria in rat hepatocytes (Sun et al, 2015) and endosomes in human macrophages (Yang et al, 2023). Notably, ZnT1 and the

---

[1]Shanghai Stomatological Hospital, School of Stomatology, Institutes of Biomedical Sciences, Fudan University, 200032 Shanghai, China. [2]Shanghai Key Laboratory of Medical Epigenetics, International Co-laboratory of Medical Epigenetics and Metabolism (Ministry of Science and Technology), Department of Systems Biology for Medicine, Fudan University, 200032 Shanghai, China. [3]Department of Pharmacology, School of Basic Medical Sciences, Fudan University, 200032 Shanghai, China. [4]Research Center for Deepsea Bioresources, 572025 Sanya, Hainan, China. [5]School of Pharmacy, Hainan Medical University, 570228 Haikou, China. [6]These authors contributed equally: Yonghui Long, Zhini Zhu, Zixuan Zhou, Chuanhui Yang, Yulin Chao. ✉E-mail: qqh@fudan.edu.cn

concomitant endosomal $Zn^{2+}$ level have been found to regulate the endocytosis of TLR4 and PD-L1 in macrophages, supporting the potential role of zinc supplement to treat inflammation-associated tumors in synergy with chemotherapy (Yang et al, 2023). Moreover, recent evidence relates somatic ZnT1 loss-of-function mutations to primary aldosteronism, the most common form of endocrine hypertension that affects 2% adults (Rege et al, 2023). Despite the physiological and pathological importance of ZnT1, the structure and transportation mechanism remain to be unveiled.

A plethora of structural studies have revealed the general dimeric architecture and different conformational states of ZnT/YiiP members (Cotrim et al, 2019). The *Escherichia coli* zinc transporter YiiP protein (ecYiiP) was crystallized in a $Zn^{2+}$-bound homologous outward-facing (OF) conformation with two TMDs splayed far apart (Lu and Fu, 2007; Lu et al, 2009). Differently, the two protomers of *Shewanella oneidensis* YiiP (soYiiP) were captured in a closely-bound homologous inward-facing (IF) conformation, via cryo-EM analysis on either helical crystals or single particles (Coudray et al, 2013; Lopez-Redondo et al, 2018, 2021). A comparison of the OF and IF conformations demonstrated a dramatic scissor-like conformational change of YiiP for $Zn^{2+}$ recognition and transportation (Lopez-Redondo et al, 2021). Recently, Bai and colleagues conducted single-particle cryo-EM analysis on the human ZnT8, and observed an intriguing heterogenous IF/OF conformation (i.e., one protomer facing outward/lumen and the other inward/cytosol) in the absence of $Zn^{2+}$ at pH 7.4, in addition to OF/OF dimers (Xue et al, 2020). Besides, during the preparation of our manuscript, the Inaba group reported the human ZnT7 with diverse structural conformations at pH 7.5, via the assistance of a stabilizing Fab (Bui et al, 2023). However, the mechanism that coupling $Zn^{2+}$ movement with proton gradient through the translocation funnel awaits further exploration.

ZnT1-mediated $Zn^{2+}$ efflux has been demonstrated a dependence on $H^+$ and/or $Ca^{2+}$ gradient (Shusterman et al, 2014; Gottesman et al, 2022). To comprehensively investigate the structural dynamics and translocation mechanism of ZnT1, we performed single-particle cryo-EM analysis on the human ZnT1 (hZnT1) directly and obtained several high-resolution density maps of sufficient quality to resolve key elements involved in $Zn^{2+}$ transportation. Akin to other zinc transporters, hZnT1 dimerizes via extensive interactions between its CTD, TMD and intriguingly the cystine-rich ECD region which exhibits a modest contribution to transport activity. In the absence of $Zn^{2+}$, ZnT1 protomers adopt predominately the OF conformation, without $Zn^{2+}$ chelated in the TMD binding site. With $Zn^{2+}$ supplementation at pH 7.5, the 2.65-Å resolution map reveals that both protomers retain the OF state with $Zn^{2+}$ coordinated at TMD sites by trihedral ligand composition instead of tetrahedral network. Interestingly, when prepared with $Zn^{2+}$ at pH 6.0, ZnT1 exhibits diverse conformations including the OF/OF and IF/IF homodimers, and the OF/IF heterodimer, all with $Zn^{2+}$-bound at TMD sites. Notably, $Zn^{2+}$ was coordinated by the typical tetrahedral network in the inward-facing state. We also determined a ZnT1 structure with $Ca^{2+}$ supplemented, however, no $Ca^{2+}$ ion could be identified. These structural snapshots, together with biochemical experiments and molecular dynamics simulation analysis, shed light on the conformational dynamics of ZnT1-mediated $Zn^{2+}/H^+$ exchange.

# Results

## Structural determination of hZnT1 in the absence of $Zn^{2+}$

Full-length hZnT1 with FLAG tag on the C termini was overexpressed in human embryonic kidney (HEK) 293 cells. Purification via affinity resin and size-exclusion chromatography of the detergent LMNG in combination with CHS produced homogenous mono-dispersed hZnT1 protein samples (Fig. 1A,B). To overcome the challenge of small molecular weight of ZnT/YiiP proteins, previous cryo-EM characterizations were facilitated by either the contrast-enhancing volta phase plate (e.g., hZnT8), or the specifically raised antibody fragments (e.g., soYiiP and hZnT7). Albeit with the successful structural determination, large fiducial references like Fabs, pro-macrobodies or legobodies often over-whelm the particle alignment and lead to relatively lower density quality on regions of interest, based on our experience and also as evidenced by the studies of SoYiiP and hZnT7 (Lopez-Redondo et al, 2021; Bui et al, 2023). Encouraged by recent advances of small fiducials like nanobody (~12 kDa) in assistance of single-particle analysis (Han et al, 2022; Robertson et al, 2022; Wang et al, 2024), we anticipated that the conserved discernible dimerized CTD domain (total ~18 kDa) of ZnT proteins would provide an adequate signal for particle alignment on its own. Indeed, good image contrast and particle distribution in thin vitrified ice yielded high-resolution reconstructions of hZnT1 (Fig. EV1A).

A global 3.48-Å resolution map was obtained for hZnT1 purified without EDTA treatment, which allows unambiguous modeling of most regions except the intracellular histidine-rich loop (residues 138–240) and the C-terminus (residues 448–507), owing to their intrinsic high mobilities (Appendix Fig. S1). ZnT1 dimerizes tightly through the intracellular CTDs, the six-helical bundle TMDs, and the extracellular cysteine-rich linker that connects TM5 and TM6 (Figs. 1C and EV2). The extracellular half of TM2 tilt towards the neighboring TM3, which creates a hydrophobic interface mainly through aromatic and aliphatic residues (Fig. 1C). These extensive interfaces contribute to an overall torpedo-shaped architecture, unlike the mushroom-shaped hZnT7, or the V-shaped ecYiiP or hZnT8 (Fig. EV3A).

Openings of both hZnT1 protomers face the extracellular space, revealing a negatively charged chamber at the TMD/ECD nexus (Fig. 1D). No obvious density was identified in the TMD $Zn^{2+}$-binding site of either protomer, indicating a $Zn^{2+}$-unbound OF/OF homodimer captured for ZnT1 (Fig. 1E). Interestingly, the sidechain of His43 on TM2 is not confined in the vicinity of Asp47/His251/Asp255, the highly conserved HD/HD tetrahedral zinc coordination network among ZnT/SLC30 family (Appendix Fig. S2). Notably, previous $Zn^{2+}$-unbound ZnT7 or ZnT8 OF/OF homodimers were determined in the presence of 1 mM zinc chelator EDTA. Considering that the cytosolic labile $Zn^{2+}$ concentration ranges from picomolar to nanomolar (Outten and O'Halloran, 2001), while the binding affinity between free $Zn^{2+}$ and TMD region is measured at the micromolar level with purified ZnT/YiiP proteins (Wei and Fu, 2005; Bui et al, 2023), our $Zn^{2+}$-unbound outward-facing ZnT1 conformation may reflect the post status when zinc ions are expelled out of the translocation passage.

hZnT1 contains a distinctive extracellular cysteine-rich region which is highly conserved among ZnT1 orthologs (Appendix Fig. S2), and not found in other ZnT family members or bacterial YiiP

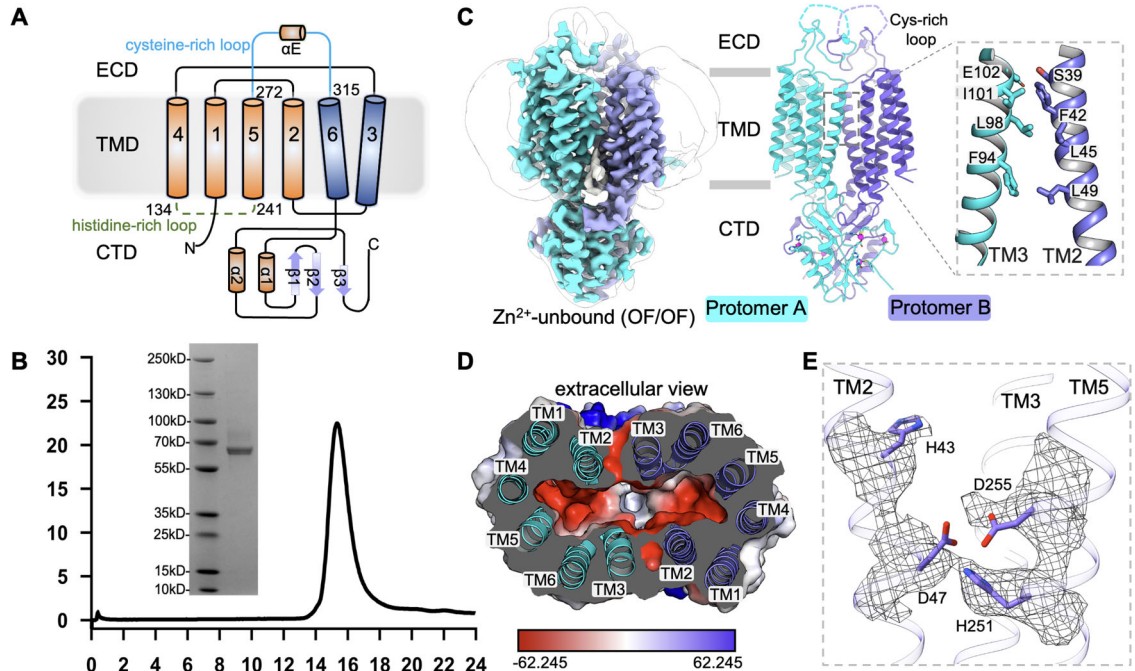

**Figure 1. Structure of Zn²⁺-free hZnT1 in the outward-facing state.**

(A) Topology diagram of the full-length wild-type hZnT1 construct used for cryo-EM. (B) Representative size-exclusion chromatography (SEC) of full-length hZnT1 and the Coomassie blue stained SDS-PAGE of the final sample concentrated. (C) Side view of cryo-EM density map of hZnT1 OF/OF homodimer, with two protomers colored cyan and blue, respectively. A Gaussian-filtered unsharpened map is sketched to indicate the extracellular density connecting transmembrane helices TM5 and TM6. Cartoon representation of hZnT1 is shown in the middle (side view), with the residues lining the TMD dimer interface shown in sticks (right). Zn²⁺ ions at CTD are shown as magenta balls. (D) Cutaway top-view of the electrostatic surface potential (negative in red, and positive in blue) reveals a negatively charged pocket in outward-facing protomers for Zn²⁺ binding. (E) Zoom-in view of the vacant Zn²⁺ chelating HD/HD network in TMD region, with side-chain density shown in the gray mesh. Source data are available online for this figure.

sequences (Appendix Fig. S3). Despite of weak local density, the main chain fits well in the unsharpened map, guided by AlphaFold2 prediction (Fig. EV2B). This region adopts a unique "lasso" shape (Fig. 2A), with a short helix contributing to the dimeric interface (Fig. EV2B). The residue registry at ECD dimeric interface was not fully sufficed, however, internal deletion of this segment (Δ281–304) modestly reduced the Zn²⁺ efflux activity (Fig. 2B), with no substantial effect on ZnT1 expression or surface localization (Fig. EV4A). Another feature in ZnT1 is the intracellular histidine-rich loop that connects TM4 and TM5 (Fig. 1A). Due to its intrinsic flexibility, modeling of this 7-histidines linker is not permitted (Fig. 2A). Interestingly, truncation of this histidine-rich loop (Δ141–210) also exhibited a small but significant effect on ZnT1-mediated Zn²⁺ transport (Fig. 2B). So far, such histidine-rich region is limited to ZnT1 and ZnT7. Notably, Bui and co-workers observed a physical interaction between its 14-histidines loop and TMD substrate binding site in the inward-facing ZnT7 structure (Bui et al, 2023). This interaction may be unique to ZnT7, as the number and distribution pattern of these histidine residues vary significantly between ZnT7 and ZnT1 (Appendix Fig. S3).

The ZnT/YiiP CTDs vary significantly in the amino acid sequences, the exact Zn²⁺ locations, and the chemical environments of binding sites (Fig. EV3B; Appendix Fig. S3), despite a conserved metallochaperone-like structural core consisting of two short α-helices and three β-strands (αββαβ-fold). In ZnT1, three CTD Zn²⁺-binding sites were identified at equilateral-triangle positions

(Fig. 2A,C), in contrast to two closely positioned Zn²⁺-sites in the YiiP or ZnT8 CTD domains, and none in ZnT7 (Lu and Fu, 2007; Coudray et al, 2013; Xue et al, 2020; Bui et al, 2023). ZnT1 $S_{CD1}$ is overlapped with $S_{CD1}$ sites found in ZnT8 or YiiP proteins, the peripheral $S_{CD2}$ is composed of residues from both protomers, and the third Zn²⁺-site ($S_{CD3}$) is close to the TMD/CTD nexus. Truncations of these Zn²⁺ sites decreased the ZnT1 activity to various extents without obvious disturbance on the protein expression and surface localization (Figs. 2B and EV4A), suggesting a potential role of these structural Zn²⁺ in the transportation. It is noted that wherein an interface Zn²⁺-site ($S_{IF}$) has been observed in ZnT8 and YiiP (Fig. EV3A), with different coordination compositions. Specifically, the Zn²⁺-chelating network of $S_{IF}$ site is mainly contributed by the intracellular loop residues connecting TM2 and TM3 (IL2-3) in YiiP, or by His137 on IL2-3 and His345 on CTD in ZnT8, while the ZnT1 $S_{CD3}$ site is composed of CTD residues Glu371 and His373, and the C-terminal extension Cys446 and Cys447. Given the transitional location and regulation of transport activity, we speculate that this $S_{CD3}$ site may function like the $S_{IF}$ site proposed in ZnT8 that facilitates Zn²⁺ trapping and subsequent transportation (Xue et al, 2020).

## Zn²⁺-bound outward-facing ZnT1 structure

To capture the Zn²⁺-bound state, we incubated ZnT1 with 1 mM ZnSO₄ at pH 7.5 and determined a global 2.65-Å resolution map of

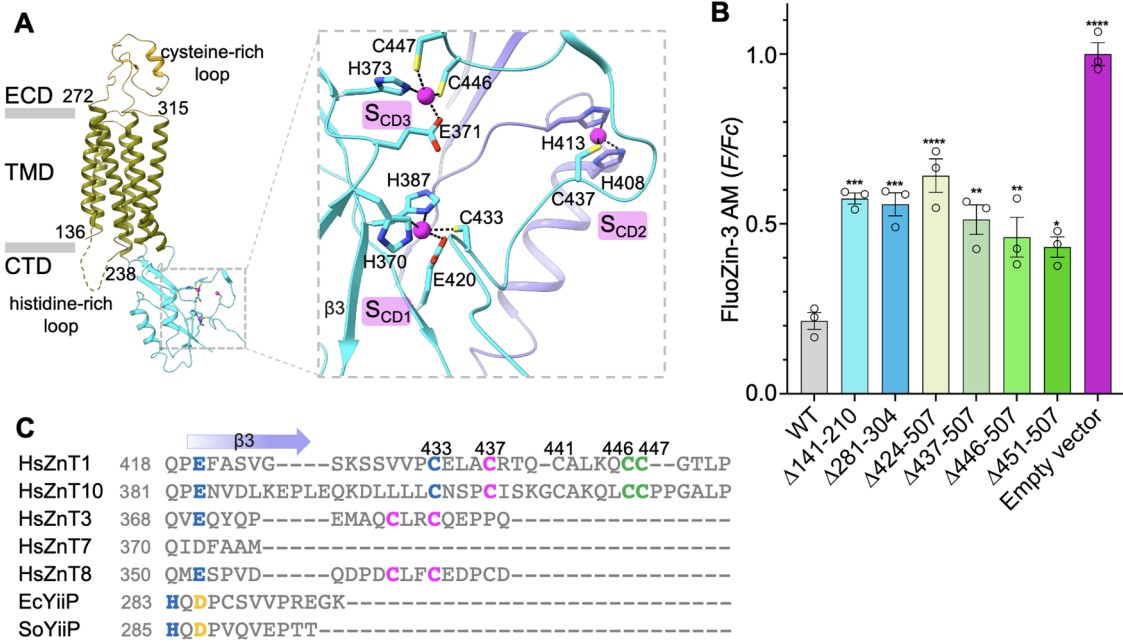

**Figure 2. Distinct elements in the regulation of hZnT1 activity.**

(A) Side view of the $Zn^{2+}$-unbound outward-facing hZnT1 protomer structure. The unmodelled intracellular histidine-rich loop is shown as a dashed line. Three $Zn^{2+}$-binding sites with distinct chemical environments are scattered in CTD, numbered as $S_{CD1}$, $S_{CD2}$ and $S_{CD3}$ after the registry of coordinating residues. (B) Intracellular FluoZin-3 AM fluorescence between WT hZnT1, and variants truncated at extracellular cysteine-rich region (Δ281–304), intracellular histidine-rich loop (Δ141–210), and C-terminus (Δ424–507, Δ437–507, Δ446–507, Δ451–507). Error bars indicate means ± SEM, $N = 3$ independent experiments, $n \geq 274$ total number of each analyzed stably transfected HEK293T cells. The fluorescence intensity ($F$) is normalized to that of control cells transfected with an empty vector ($Fc$). Significance was analyzed by one-way ANOVA with Turkey post hoc test. *$P < 0.05$, **$P < 0.01$, ***$P < 0.001$, ****$P < 0.0001$, ns = non-significant. $P = 0.0001$, 0.0002, $1.52 \times 10^{-5}$, 0.001, 0.0062, 0.0178, $3.14 \times 10^{-9}$. (C) Sequence alignment of the C-terminal extension regions of representative ZnTs, hZnT1 (Q9Y6M5), hZnT10 (Q6XR72), hZnT3 (Q99726), hZnT7 (Q8NEW0), hZnT8 (Q8IWU4), and two batererial YiiP homologs, EcYiiP (P69380) and SoYiiP (Q8E919). Residues involved in different coordination sites are colored accordingly (blue, $S_{CD1}$; magenta, $S_{CD2}$; green $S_{CD3}$; orange, shared by $S_{CD1}$ and $S_{CD2}$).

C1 symmetry (Figs. 3A and EV1B). Local resolution analysis indicated density quality better than 2.5-Å for TMD $Zn^{2+}$ sites (Fig. EV1B), which showed clear signs for $Zn^{2+}$ ions present at both outward-facing TMD translocation funnels (Fig. 3B). This high-quality density permits confident modeling for most regions. The overall $Zn^{2+}$-bound structure is nearly identical to the $Zn^{2+}$-unbound OF/OF homodimer, with a Cα RMSD of 0.5 Å.

Similar to the $Zn^{2+}$-unbound OF/OF ZnT1 structure, three $Zn^{2+}$-binding $S_{CD}$ sites were also identified in $Zn^{2+}$-bound OF/OF ZnT1 (Fig. 3A), without an additional TMD/CTD interface $S_{IF}$ site as seen in hZnT8 or bacterial YiiP structures when incubated with zinc (Fig. EV3A). The two $Zn^{2+}$-bound protomers are well aligned. Notably, a short segment of TM2 extracellular half (residues 41–46) in both protomers adopts a stretched conformation (Figs. 3B and EV2C). As a result, zinc is chelated by three residues Asp47, His251 and Asp255, with His43 swinging away from the ion, which resembles the $Zn^{2+}$-unbound apo state (trihedral-network, Fig. 3C).

## Conformational dynamics at low pH

ZnT1 has been characterized as a $Zn^{2+}/H^+$ exchanger (Shusterman et al, 2014; Cotrim et al, 2021), a common transport mechanism among ZnTs and bacterial YiiP proteins. Aside from surface expression, ZnT1 can be found in cytoplasmic compartments such as mitochondria (Sun et al, 2015) and endosomes (Yang et al,

2023), implying similar scenarios for pH-driven $Zn^{2+}$ transport as other cytoplasmic ZnTs. Recent structural analysis identified distinct conformations of hZnT8, hZnT7, ecYiiP or soYiiP, i.e., the outward/lumen and the inward/cytosol-facing states (Lu and Fu, 2007; Lopez-Redondo et al, 2021; Xue et al, 2020; Bui et al, 2023), which were captured under pH 7.0 ~ 7.5 conditions. Stokes and colleagues also observed an intermediate between inward- and outward-facing states when removing $Zn^{2+}$ with EDTA, suggesting the role of zinc in driving the transport cycle of soYiiP protein (Lopez-Redondo et al, 2021). These studies provide important insights into the transport dynamics of ZnT/YiiP family; however, it remains unclear how $H^+$ is coupled to $Zn^{2+}$ movement.

To address this question, we prepared hZnT1 cryo-EM sample in a low pH buffer (50 mM MES, pH 6.0) with 1 mM $ZnSO_4$. Surprisingly, we obtained three major conformations with $Zn^{2+}$-bound at TMD sites after extensive particle classification: the OF/OF homodimer, the IF/IF homodimer, and the OF/IF heterodimer (Figs. 4A and EV1C). The overall OF/OF structure obtained at pH 6.0 is nearly identical to the aforementioned OF/OF homodimers determined at pH 7.5.

Although the two protomers in all three conformations remain dimerized via CTD, TMD and ECD, the interface constituents vary substantially. For instance, the TMD dimerization in IF/IF homodimer is mainly mediated by TM3 helices, as the TM2 extracellular apex swings away from TM3 of the neighboring

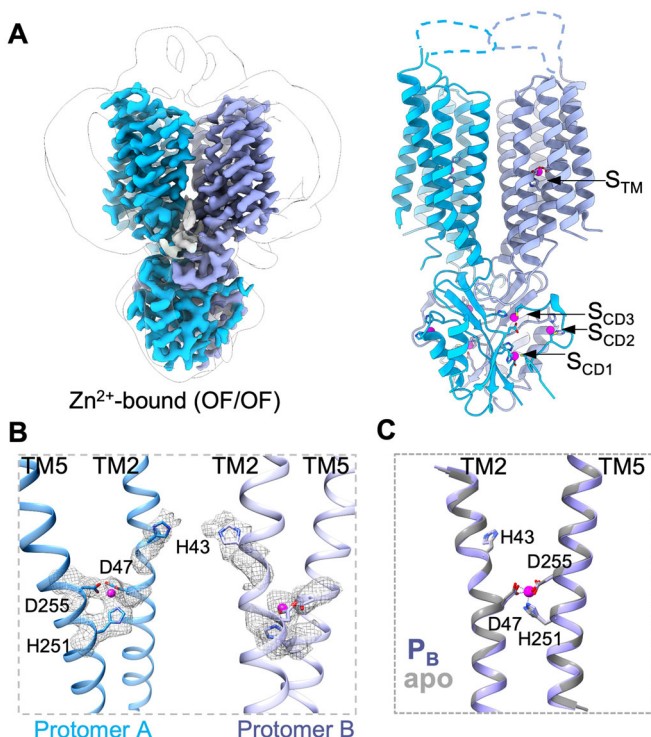

**A**

Zn$^{2+}$-bound (OF/OF)

S$_{TM}$

S$_{CD3}$
S$_{CD2}$
S$_{CD1}$

**B**

TM5  TM2   TM2  TM5

H43

D47

D255

H251

Protomer A    Protomer B

**C**

TM2        TM5

H43
D255

D47
H251

P$_B$
apo

Figure 3.  Zn$^{2+}$-bound substrates of hZnT1 in outward-facing conformation.

(A) Side view of Zn$^{2+}$-bound hZnT1 OF/OF dimer 2.65-Å resolution density map overlaid within a Gaussian-filtered unsharpened map. Zn$^{2+}$-binding sites are labeled accordingly on right panel. (B) Two protomers with Zn$^{2+}$ coordinated by three residues at the TMD region. (C) Overlay of the Zn$^{2+}$-bound protomer (blue) with one Zn$^{2+}$-unbound protomer of OF/OF homodimer (gray).

protomer (Fig. 4B), compared to that formed by two pairs of TM2-TM3 interactions in OF/OF homodimer (Fig. 1C). Of note, TM2 segment (residues 41–46) in Zn$^{2+}$-bound IF state adopts the canonical helix shape (Fig. EV2D), with Zn$^{2+}$ chelated by a typical tetrahedral coordination network (His43/Asp47/His252/Asp255). Interestingly, the heterodimer TMD interface is formed by TM3 of the inward-facing protomer with TM2 and TM3 of the outward-facing protomer, with shared contributing residues observed in both OF/OF and IF/IF homologous dimeric interfaces (Fig. 4C), indicating an intermediate between the OF/OF and IF/IF states. Moreover, the interaction of extracellular "lasso" regions in IF/IF homodimer is slightly more stable than that of OF/IF heterodimer and OF/OF homodimer, evidenced by the better-resolved density compared at the same map contour levels (Fig. EV3C). Structural alignment of the inward-facing and outward-facing TMDs revealed that the TMs 1, 2, 4, and 5 move concertedly against the TM3/6 scaffold (Fig. 4D), similar to the movement observed in the YiiP, ZnT7, and ZnT8 structures.

## hZnT3 at inward-facing Zn$^{2+}$-bound state

To validate if the hZnT1 conformational dynamics obtained at low pH is generally adopted by ZnTs, we sought to characterize a second ZnT family member, the synaptic vesicle-specific zinc transporter ZnT3 (Palmiter et al, 1996). Vesicular zinc in the brain

modulates neuronal development, synaptic plasticity and cognitive function. ZnT3$^{-/-}$ mice suffered from impaired contextual discrimination and spatial working memory (Adlard et al, 2010; Martel et al, 2011).

We prepared hZnT3 in the same low pH buffer as for hZnT1, and successfully obtained a 3.14-Å resolution V-shaped map which permitted the modeling of an IF/IF homodimer (Fig. EV1E), with zinc ions identified at TMD binding sites (Fig. 5A). Extra densities were also observed, including an extended density lying along the cleft between the two protomers, and branched densities wedged in the TMDs (Appendix Fig. S4A). We tentatively assigned a phosphoethanolamine molecule (PE) for the elongated density (Appendix Fig. S4B), and the lauryl maltose neopentyl glycol (LMNG) detergent for the branched density (Appendix Fig. S4C). The artificial engagement of LMNG with TMDs is unexpected, especially it is situated right beneath the S$_{TM}$ Zn$^{2+}$-binding site. This interaction may lock TMD in the inward-facing state and could partially explain why we could not capture different conformations driven by protons at low pH condition.

Although the V-shaped architecture with two splayed TMDs is similar to its closest paralog hZnT8, some specific features are observed only in hZnT3. A Zn$^{2+}$-binding S$_{IF}$ site was identified at the TMD/CTD nexus in a recent ZnT8 structure, with Zn$^{2+}$ coordinated by two histidine residues (His137 and His345). In contrast, this interface Zn$^{2+}$ site is not preserved in hZnT3, and the His345 is replaced by a serine (Ser363) at an equivalent position on CTD (Fig. 5B). Moreover, the aforementioned lipid-like densities stuffed between ZnT3 TMDs were not observed in ZnT8 (Xue et al, 2020). Mutation of the residue Arg286 close to the PE head group modestly reduced the Zn$^{2+}$ transport activity, without significant alterations to ZnT3 expression or localization (Fig. EV4B).

A similar sample preparation procedure allows rational comparison between ZnT1 and ZnT3, which belong to two different subgroups of ZnT family. Despite different dimerization patterns observed in the tightly bound torpedo-shaped hZnT1 and the splayed V-shaped hZnT3 IF/IF homodimers, their TMDs actually can be well aligned (Fig. 5C), suggesting a conserved transport mechanism among ZnT/YiiP family. The inserted LMNG molecule at TMD site is unique to ZnT3, albeit with unknown reason, implying potential modulation on ZnT3 activity.

## Transport mechanism

ZnT/YiiP mediated Zn$^{2+}$ transport is mainly powered by proton gradient. To provide mechanistic insight into how ZnT members including ZnT1 transport Zn$^{2+}$, we performed atomistic ensemble molecular dynamics simulations (Figs. 6A and EV5). In the outward-facing state of ZnT1, Zn$^{2+}$ was modeled to be coordinated by His43/Asp47/His251/Asp255 residues which remained stable in the non-protonated tetrahedral network throughout 1 μs duration (Fig. EV5A; Movie EV1). When His43 was protonated, His43 sidechain became mobile and distant from Zn$^{2+}$, and TM2 segment stretched (Movie EV2). We speculated that the partial positive charge conferred by protonation causes the weakly confined His43 to become repulsive against Zn$^{2+}$ and significantly volatile, which introduces perturbation in the surrounding chemical environment and transforms into concerted movement of TM2 segment. Previous characterization suggested a selective role of this histidine in ZnT5 or ZnT8 proteins favoring Zn$^{2+}$ over Cd$^{2+}$ (Hoch et al, 2012). The substitution of His43

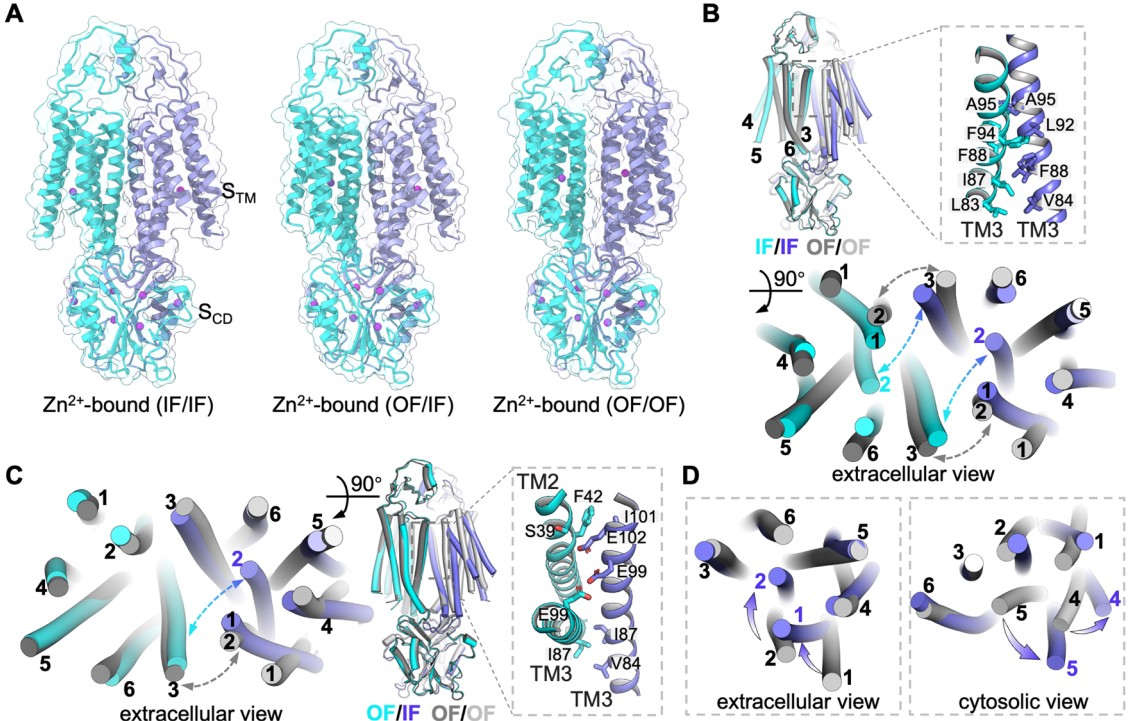

**Figure 4. Conformational dynamics of hZnT1 under low pH condition.**

(A) Side view of three $Zn^{2+}$-bound hZnT1 conformations obtained at pH 6.0. (B) Structural alignment between the $Zn^{2+}$-bound IF/IF (cyan/blue) and OF/OF (dark/light gray) homodimers. The TMD dimeric residues are shown as sticks on the right, with the rest elements hidden for clarity. The distance variations between the extracellular apexes of TM2 and TM3 in the IF/IF and OF/OF states are indicated by dashed arrows on bottom. (C) Overlay of the $Zn^{2+}$-bound OF/IF heterodimer (cyan/blue) with OF/OF homodimer (dark/light gray). The distance variations between the extracellular apexes of TM2 and TM3 are indicated by dashed arrows (left), and the intermediate dimeric interface is shown on the right. (D) TMD comparison of the inward-facing protomer (blue) with the outward-facing protomer (gray) viewed from the extracellular side (left) and cytosolic side (right). TM movements are indicated by arrows.

with alanine substantially reduced ZnT1 activity by 70%, while either H43D or H43N mutant retained the $Zn^{2+}$ transport activity (Fig. 6B). Protonation on both His43 and His251 did not alter the positioning of $Zn^{2+}$ (Movie EV3), suggesting additional protonation is required for zinc release from translocation passage. Indeed, when Asp47 and Asp255 were further protonated, the zinc coordination network was disrupted and $Zn^{2+}$ was released (Fig. 6A; Movies EV4, EV5, and EV6).

To investigate the dynamics of $Zn^{2+}$ reception, initial MD model was generated by removing $Zn^{2+}$ from $Zn^{2+}$-bound IF/IF ZnT1 structure. Interestingly, the His251 that close to intracellular entry site appeared less bound to the coordination site before $Zn^{2+}$ was accepted. When zinc entered the translocation funnel, the tetrahedral network gradually became stable, preparing for subsequent conformation transition (Fig. EV5B; Movie EV7). We also simulated the $Zn^{2+}$ recognition process for the inward-facing ZnT3 in a similar setting. Likewise, the His238 (which corresponds to His251 in ZnT1) was not confined in the tetrahedral coordination network before $Zn^{2+}$ entered. When $Zn^{2+}$ get in the position, the His108/Asp112/His238/Asp242 network was built up and ready for $Zn^{2+}$ recognition and translocation (Fig. EV5C).

Recent studies also suggested a critical role for $Ca^{2+}$ in the ZnT1-mediated $Zn^{2+}$ efflux (Gottesman et al, 2022; Shusterman et al, 2014). To investigate the molecular mechanism of $Zn^{2+}/Ca^{2+}$ exchange, we performed cryo-EM analysis on the ZnT1/$Ca^{2+}$

sample, which was prepared similarly to the $Zn^{2+}$-bound ZnT1 samples. Unexpectedly, no obvious additional extra density corresponding to $Ca^{2+}$ could be observed in the TMD region of a 3.52-Å map (Fig. EV1D). This leaves the question of whether $Ca^{2+}$ may bind to ZnT1 in the same way as the cognate substrate $Zn^{2+}$ remains open.

Incorporating ZnT1's dynamic conformations under low pH condition and the general alternating access mechanism, we propose the rocking-bundle working model for ZnT1 (Fig. 6C). The cytosolic zinc accesses the substrate binding site when ZnT1 is in an inward-open conformation. When zinc is recognized by the tetrahedral group HD/HD, the transporter switches the TM bundles (TMs 1, 2, 4, and 5) towards an outward-facing state. Protonation on key residues including His43 and His251 allows zinc release, and possibly resets ZnT1 back to the inward-open state, for another transport cycle.

## Discussion

ZnT/SLC30 transporters are major players in regulating zinc homeostasis, by harnessing the proton gradient to mobilize the divalent cationic zinc across membranes. Here we determined four high-resolution structures of the plasma-specific hZnT1 and the synaptic vesicle-specific hZnT3. Particularly, we obtained different

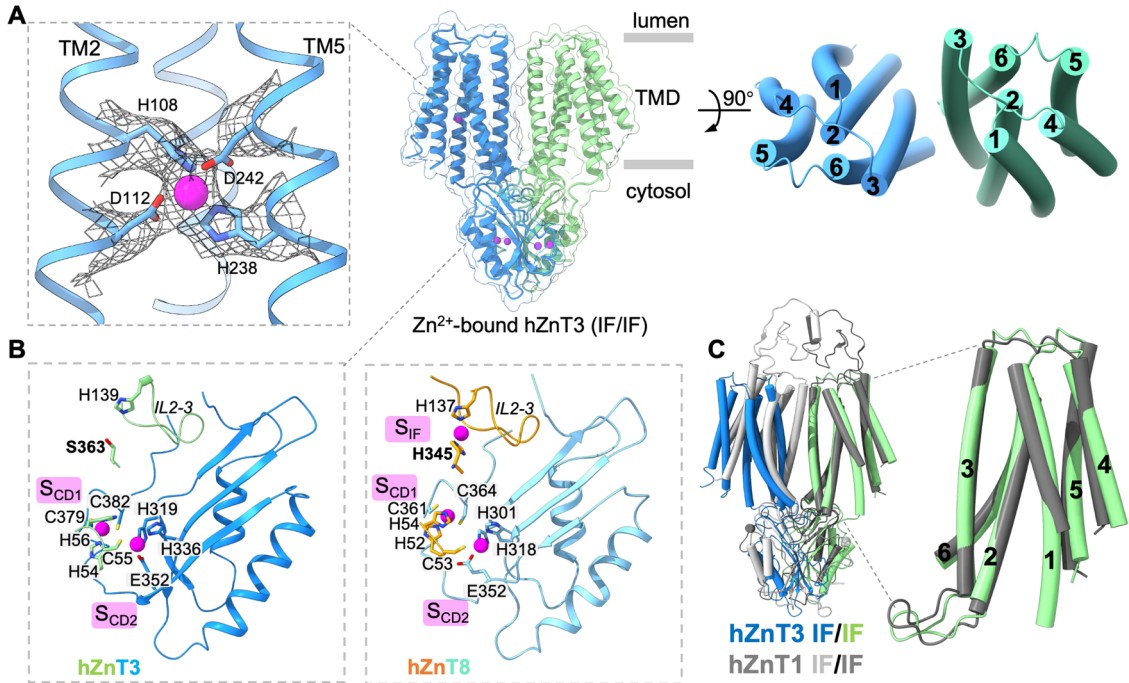

**Figure 5. Zn²⁺-bound hZnT3 at inward-facing conformation.**

(A) Overall structure of Zn²⁺-bound hZnT3 inward-facing homodimer obtained at pH 6.0. Left, zoom-in view of TMD Zn²⁺-binding site with EM density of key residues shown as mesh. Right, dimeric architecture viewed from lumen side. (B) Comparison of CTD regions between hZnT3 (left) and its closest paralog hZnT8 (right). Two $S_{CD}$ Zn²⁺-binding sites are conserved with similar coordinating residues, while the TMD/CTD interface $S_{IF}$ site observed in hZnT8 is not preserved in hZnT3. (C) Overlay of the Zn²⁺-bound hZnT3 (blue/green) and hZnT1 (dark/light gray) IF/IF homodimers. The inward-facing TMDs are well aligned (right).

conformations of hZnT1 at low pH conditions. Supported by molecular dynamics simulation and biochemical experiments, these observations provide insights into the mechanisms underlying the Zn²⁺/H⁺ exchange process.

Despite amassing a plethora of biochemical and structural knowledge, we are still unable to attribute a physiological function to the CTD as a whole, or better, because they are likely to differ, to individual ZnT/YiiP members. Different numbers and locations of Zn²⁺-binding site have been characterized for ZnT/YiiP members. Except that ZnT7 has no CTD Zn²⁺ site, there are usually two Zn²⁺ sites located at CTD region. Intriguingly, three Zn²⁺-binding sites are identified at hZnT1 CTDs, with the third site close to the TMD/CTD nexus. It is recognized that ZnT8 and YiiP contain an $S_{IF}$ site at the TMD/CTD interface. Notably, this $S_{IF}$ site is not found in hZnT3, which belongs to the same subgroup as ZnT8. Our study and others have characterized a role for the CTD Zn²⁺-binding sites in regulating transport activity. These observations suggest that ZnT/YiiP members may adopt different strategies to enrich local zinc concentrations and promote translocation, given the picomolar cytosolic free zinc levels.

It is worth discussing the potential modulation of lipids on ZnT transporters. In the current study, we observed lipid-like densities located between hZnT3 TMDs. Although we could not determine the exact identity at this moment, our cellular assay suggested a possible regulatory role for this lipid-like molecule. In line with our observation, the transport activity of ZnT8, the closest paralog of ZnT3, was also shown to be tuned by anionic lipids (Merriman et al, 2016). Notably, the exogenous HEK293 overexpression system

may not fully recapitulate the synaptic vesicle environment where hZnT3 locates physiologically, therefore the lipid identity and potential modulation await future exploration in a more natural context, for instance the neurons.

During the revision of our manuscript, Sun and co-workers determined a cryo-EM structure of ZnT1 in the Apo state at 3.4-Å resolution and characterized the Ca²⁺ dependence on ZnT1-mediated Zn²⁺ transport in both cellular assays and in proteoliposomes (Sun et al, 2024). However, the absence of a Ca²⁺-bound structure limits our mechanistic understanding of Ca²⁺-coupled Zn²⁺ transport, necessitating further exploration.

In conclusion, the significant structural changes of ZnT1 captured in the presence of Zn²⁺ and low pH circumstances, together with biochemical analysis and MD simulations, provide insight into the ZnT1-mediated Zn²⁺ efflux.

## Methods

### Reagents and tools table

| Reagent/resource | Reference or source | Identifier or catalog number |
|---|---|---|
| **Experimental models** | | |
| Expi293F cells | ThermoFisher Scientific | Cat # A14527 |
| HEK293T cells | ATCC | Cat # CRL-11268 |

| Reagent/resource | Reference or source | Identifier or catalog number |
|---|---|---|
| **Recombinant DNA** | | |
| pCDNA3.1-ZnT1-TEV site-Flag-mCherry | This paper | N/A |
| pCDNA3.1-ZnT3(50-388)-3C site-Flag | This paper | N/A |
| pCDH-ZnT1-3C site-Twin Strep tag-HA tag-mCherry and mutants | This paper | N/A |
| pCDH-ZnT3-3C site-Twin Strep tag-Flag tag-mCherry and mutants | This paper | N/A |
| **Antibodies** | | |
| Rabbit anti-HA antibodies | Cell Signaling Technology | Cat # 3724 |
| Rabbit anti-β-actin | Proteintech | Cat # 20536-1-AP |
| Mouse anti-DYKDDDK antibodies | ABmart | Cat # M20008 |
| Rabbit anti-sodium potassium ATPase | ABmart | Cat # M40013s |
| HRP-conjugated goat anti-rabbit antibody | Signalway Antibody | Cat # L3012 |
| HRP-conjugated goat anti-mouse antibody | Signalway Antibody | Cat # L3032 |
| **Chemicals, enzymes, and other reagents** | | |
| FluoZin-3 | ThermoFisher Scientific | Cat # F24195 |
| 1,10-Phenanthroline | Sigma | Cat# 131377 |
| Sodium Butyrate | Aladdin | Cat # S102956 |
| n-dodecyl-β-D-maltoside | Anatrace | Cat # D310 |
| cholesteryl hemisuccinate | Anatrace | Cat # CH210 |
| lauryl maltose neopentyl glycol | Anatrace | Cat # NG310 |
| 3×FLAG peptide | Smart-Lifesciences | Cat # SLR01002 |
| FreeStyle 293 Expression Medium | Gibco | Cat # C12338018 |
| DMEM basic Medium | Gibco | Cat # C11995500BT |
| Fetal Bovine Serum | Excell | Cat # FSP500 |
| TEV protease | Purification by our lab | N/A |
| **Software** | | |
| GraphPad Prism v9.3.1 | GraphPad Software | https://www.graphpad.com/scientific-software/prism |
| IBM SPSS Statistics 20 | IBM SPSS software | https://www.ibm.com/spss |
| ImageJ | ImageJ software | https://imagej.net/software/imagej/ |
| Origin 9.0 | OriginLab Corp. | https://www.originlab.com |
| RELION | | http://www2.mrc-lmb.cam.ac.uk/relion |

| Reagent/resource | Reference or source | Identifier or catalog number |
|---|---|---|
| cryoSPARC | Punjani et al, 2017 | https://cryosparc.com |
| Chimera | | https://www.cgl.ucsf.edu/chimera; RRID:SCR_004097 |
| ChimeraX | Goddard et al, 2018 | https://www.rbvi.ucsf.edu/chimerax/ |
| PyMol | Schrödinger | https://pymol.org/2; RRID:SCR_000305 |
| COOT | Emsley et al, 2010 | https://www2.mrc-lmb.cam.ac.uk/ personal/pemsley/coot; RRID:SCR_014222 |
| PHENIX | Adams et al, 2010 | https://www.phenix-online.org |
| **Other** | | |
| Anti-DYKDDDDK(FLAG) affinity resin | Smart-Lifesciences | Cat # SA042100 |
| Amicon® Ultra Centrifugal Filter, 100 kDa | MilliporeSigma | Cat # UFC910096 |
| Superose 6 Increase10/300 GL | GE Healthcare | Cat# 29091596 |
| Quantifoil R 1.2/1.3 grid Au300 | Quantifoil | Cat# Q37572 |
| Blotting Paper | Ted Pella | Cat # 47000-100 |
| PDL (poly-D-lysine)-coated 20 mm glass-bottom dishes | Sorfa Life Science | Cat # 201100 |

## Cloning and protein expression

A cDNA encoding the human full-length, wild-type ZnT1 (Uniprot: Q9Y6M5) was synthesized with codon-optimization for expression in human cells. The sequence was subcloned into a pcDNA3.1(−) vector, with a C-terminal tag containing a short GS linker, a 3C protease restriction site, followed by a EFSR-LEEELRRRTEPGS linker, TEV protease restriction site, FLAG tag and a monomeric mCherry. Recombinant expression of ZnT1 was carried out in HEK293F cells. Briefly, one-milligram plasmid of ZnT1 and 3 mg of polyethylenimine (Bioon) were mixed in 100 mL of medium for 15 min at RT before being added to 1 L of cell culture at a density of $2 \times 10^6 \, mL^{-1}$, containing 10 mM sodium butyrate. Cells were harvested after 60 h by centrifugation at $1500 \times g$ for 15 min, washed once with PBS buffer, flash-frozen in liquid nitrogen, and stored at −80 °C until use.

## ZnT1 purification

Cell pellets from 1 L of culture were thawed at 37 °C and washed in a hypotonic buffer on ice (50 mM HEPES-Na, pH 7.5, 5% glycerol). Pellets were re-suspended in buffer A (50 mM HEPES-Na, pH 7.5, 150 mM NaCl, 5% glycerol) containing 1% (w/v) n-dodecyl-β-D-maltoside (DDM, Anatrace), 0.1% (w/v) cholesteryl hemisuccinate (CHS, Anatrace), 5 mM phenylmethylsulfonyl fluoride (PMSF) and 1× protease inhibitor cocktail. The re-suspended cell pellets were

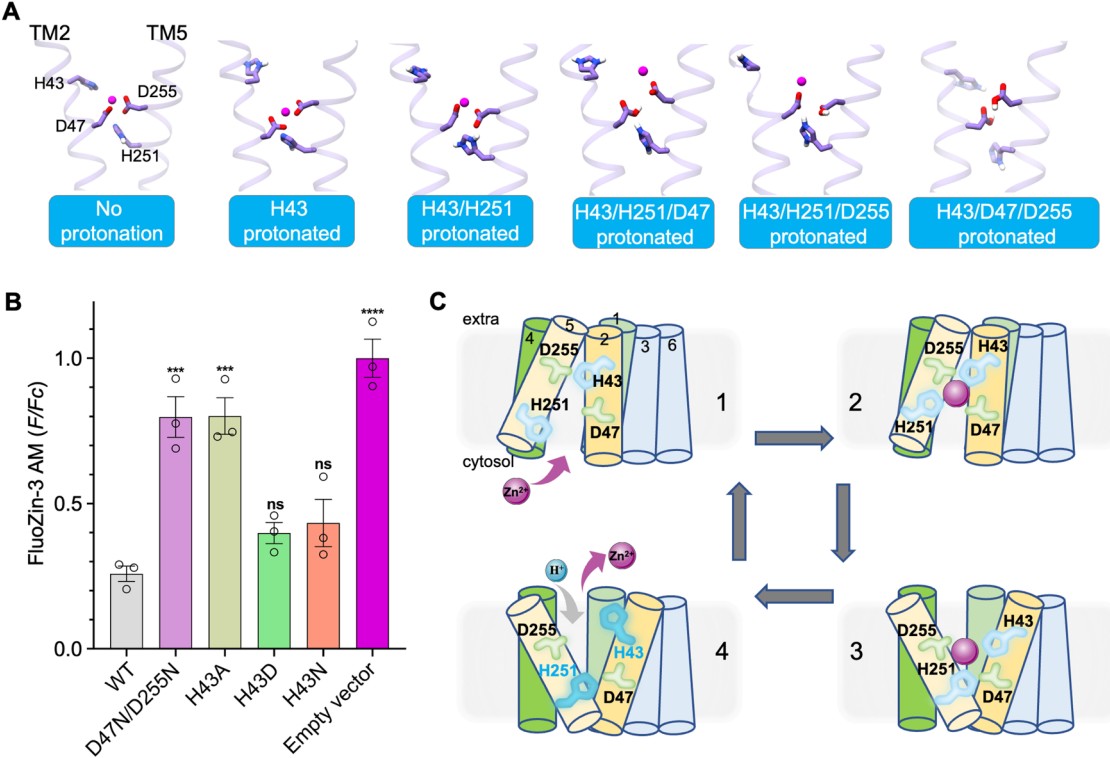

**Figure 6. Proposed working model.**

(A) Molecular dynamic snapshots of hZnT1 in the outward-facing state with indicated protonation states. For simplicity, only $Zn^{2+}$-coordinating residues on TM2 and TM5 from one protomer are shown. Zinc ion is shown as pink ball. (B) Intracellular FluoZin-3 AM fluorescence between WT hZnT1 and indicated mutants. Error bars indicate means ± SEM, $N = 3$ independent experiments, $n \geq 421$ total number of each analyzed stably transfected HEK293T cells. The fluorescence intensity ($F$) is normalized to that of control cells transfected with empty vector ($Fc$). Significance was analyzed by one-way ANOVA with Turkey post hoc test. ***$P < 0.001$, ****$P < 0.0001$, ns = non-significant. $P = 0.0004, 0.0004, 0.5882, 0.3716, 1.9 \times 10^{-5}$. (C) Rocking-bundle movement of hZnT1-mediated $Zn^{2+}$ efflux. In state 1, the transporter faces the cytosol, with TMD site vacant for zinc reception. A conformational transition is induced by zinc binding (state 2) towards the outward-facing state (state 3). Release of zinc is facilitated by the protonation of the paired histidine residues and possibly the two aspartate residues (state 4). Source data are available online for this figure.

dounced by a glass homogenizer for 0.5 h and stirred gently at 4 °C for 2.5 h in lysis buffer. After agitation, insoluble fractions were removed by centrifuging at 14,000 rpm at 4 °C for 40 min. The supernatant was incubated with 1 mL anti-DYKDDDDK(FLAG) affinity resin (Smart-Lifesciences) by agitation for 4 h at 4 °C. The resin was packed into a gravity column and washed with 10 column volume (CV) of 0.1% DDM and 0.01% CHS in buffer A. Then, the resin was successively washed with 10 CVs of buffer B (buffer A supplemented with 1% lauryl maltose neopentyl glycol (LMNG, Anatrace), 0.1% CHS), buffer C (buffer A supplemented with 0.1% LMNG, 0.01% CHS), buffer D (buffer A supplemented with 0.01% LMNG, 0.001% CHS) and buffer E (buffer A supplemented with 0.001% LMNG, 0.0001% CHS). ZnT1 was eluted with buffer (buffer E supplemented with 0.4 mg/ml 3×FLAG peptide). The elution fractions were treated with TEV protease digestion overnight at 4 °C. The digestion products were concentrated and further purified by size-exclusion chromatography (SEC) on a Superose6 10/300 GL column (GE Healthcare) in buffer containing 0.001% LMNG, 0.0001% CHS, 50 mM HEPES-Na, pH 7.5, and 150 mM NaCl, with or without 1 mM ZnSO$_4$. Peak fractions were concentrated to 1 mg/ml using a 100-kDa cut-off concentrator for cryo-EM sample preparation.

To prepare ZnT1 at low pH condition, the expression and purification were carried out similarly as described above, and the SEC buffer was switched to 0.001% LMNG, 0.0001% CHS, 50 mM MES pH 6.0, 150 mM NaCl, and 1 mM ZnSO$_4$. For ZnT1/Ca$^{2+}$ sample preparation, the SEC buffer containing 0.001% LMNG, 0.0001% CHS, 50 mM Tris-HCl pH 8.0, and 150 mM NaCl, with 1 mM CaCl$_2$ was used.

## ZnT3 expression and purification

Human ZnT3 lacking the N-terminal 49 residues followed by a C-terminal short SD linker, 3C protease restriction site, a twin-strep tag, and FLAG tag, was cloned into the pcDNA3.1(-) vector. HEK293F cells were transfected with the ZnT3 plasmids at a density of $2 \times 10^6$ cells per mL and 10 mM sodium butyrate was added. Cells were harvested after 58 h and re-suspended in lysis buffer (50 mM HEPES-Na, pH 7.5, 150 mM NaCl, 5% glycerol, 5 mM PMSF and 3 μg/ml protease inhibitor). The re-suspended cells were lysed mechanically with a Dounce tissue grinder and agitated at 4 °C for 3 h in lysis buffer containing 1% DDM, 0.1% CHS. After agitation, the supernatant was collected after centrifugation at 12,000 rpm at 4 °C for 40 min and incubated with

anti-Flag affinity resin by agitation for 3 h. Then the resin was collected on a gravity column and the supernatant was incubated with new anti-Flag affinity resin by agitation for 3 h again. The resin was washed with buffer containing 0.1% DDM, 0.01% CHS, 50 mM HEPES-Na, pH 7.5, 500 mM NaCl, and 5% glycerol. Then the resin was gradient displaced into buffer containing 0.001% LMNG, 0.0001% CHS, 50 mM HEPES-Na, pH 7.5, 500 mM NaCl and 5% glycerol, and eluted with buffer E. The elution was concentrated and further purified by size-exclusion chromatography on a Superose6 10/300 GL column (GE Healthcare) in buffer containing 0.001% LMNG, 0.0001% CHS, 50 mM MES, pH 6.0, 1 mM $ZnSO_4$, and 150 mM NaCl. Peak fractions concentrated to 2.8 mg/ml for cryo-EM sample preparation.

## Stable cell lines and cell culture

Human ZnT1 followed by a C-terminal short SD linker, twin-strep tag and HA tag, was cloned into the pCDH vector. Human ZnT3 followed by a C-terminal short SD linker, twin-strep tag and FLAG tag, was cloned into the pCDH vector. To generate mCherry-tagged stable ZnT1 or ZnT3 cell lines, HEK293T cells were transfected with the WT or mutant plasmids (pCDH with target genes, pMD2.G and psPAX2) according to the manufacturer's procedures (System Biosciences). Cells transfected with an empty vector are used for control. Cells were selected for 10 days using 3 μg/mL puromycin and maintained in 6-cm cell culture dishes (NEST) at 37 °C and 5% $CO_2$ in an incubator.

## $Zn^{2+}$ transport assay

The day before $Zn^{2+}$ transport assays, stably transfected ZnT1 or ZnT3 cells were seeded onto PDL (poly-D-lysine)-coated 20-mm glass-bottom dishes (Zhejiang Sorfa Life Science Research Co., Ltd.) at a density of $6 \times 10^4$ cells per dish, maintained in 1 mL Dulbecco's Modified Eagle Medium (DMEM basic, Gibco, cat. C11995500BT) containing 10% FBS (Excell) for 24 h. Notably, DMEM basic medium contains 1.8 mM calcium. Cells were then washed with PBS, treated with 60 μM $ZnSO_4$ and 1 μM membrane-permeable FluoZin-3 AM (ThermoFisher Scientific, cat. F24195) in DMEM for 50 min at 37 °C. Cells were washed three times with the buffer (20 mM HEPES-Na pH 7.4, 125 mM KCl, 5 mM NaCl, 10 mM Glucose, and 10 mM phenanthroline) for microscopy imaging. Live cells were then immediately monitored by a Leica SP8 LSCM+ laser scanning confocal microscope. FluoZin-3 AM was excited with a 488-nm laser line, and the emitted light measurement ranged from 500 to 570 nm. mCherry was excited at 552 nm, and the emitted light measurement ranged from 562 to 632 nm. After imaging, FluoZin-3 AM fluorescence intensities in cells transfected with empty vector were analyzed in ImageJ to take the average values as background control ($Fc$). Cells expressing ZnT1 or ZnT3 variants at similar levels among all groups were selected, with FluoZin-3 AM signal measured and averaged as ($F$) and normalized against control ($Fc$). Relative efflux capacity for ZnT1 or ZnT3 is expressed as $F/Fc$. Data analysis was performed using GraphPad Prism v9.3.1.

## Western blotting

Stable polyclonal HEK293T cells expressing HA-tagged hZnT1 or mutants were washed three times with PBS. They were then lysed by mixing with SDS loading buffer and loaded onto a 10% (w/v) SDS-PAGE gel. Rabbit anti-HA antibodies (Cell Signaling Technology) were used to detect ZnT1. Rabbit anti-β-actin (Proteintech) was used as a loading control. HRP-conjugated goat anti-rabbit antibody (Signalway Antibody) was used as the secondary antibody.

Stable polyclonal HEK293T cells expressing FLAG-tagged hZnT3 or mutants were washed three times with PBS. They were then lysed by mixing with SDS loading buffer and loaded onto a 10% (w/v) SDS-PAGE gel. Mouse anti-FLAG antibodies (ABmart) were used to detect HsZnT3. Rabbit anti-sodium potassium ATPase (ABmart) was used as a loading control. HRP-conjugated goat anti-mouse antibody (Signalway Antibody) and HRP-conjugated goat anti-rabbit antibody (Signalway Antibody) was used as the secondary antibody.

The membrane was visualized with ChemiDoc MP Imaging System (Bio-Rad) using High-sig ECL Western Blotting Substrate (Tanon).

## Cryo-EM sample preparation and data collection

Quantifoil Au 1.2/1.3 (300 mesh) grids were glow-discharged (10 mA for 50 s) in an PELCO easiGlo instrument (Ted Pella), applied 2.5 μl of purified hZnT1 protein or hZnT3, blotted with filter paper for 3 s (100% humidity at 4 °C) and vitrified in liquid ethane on a Vitrobot Mark IV (FEI).

All grids except the $Zn^{2+}$-bound ZnT1 grid were loaded in a Titan Krios cryo-electron microscope (ThermoFisher) operated at 300 kV with a 50-μm condenser lens aperture, spot size 5, magnification at 105,000× (corresponding to a calibrated sampling of 0.832 Å per physical pixel), and a K3 direct electron device equipped with a BioQuantum energy filter operated at 20 eV (Gatan). Micrographs were collected automatically using the Serial EM software with the K3 detector operating in counting mode at a recording rate of 10 raw frames per second and a total exposure time of 2 s, yielding 40 frames per stack and a total dose of 52 e-/Å². 

For $Zn^{2+}$-bound ZnT1, the data were collected on a Titan Krios G4 cryo-electron microscope operated at 300 kV, equipped with a Falcon G4i direct electron detector with a Selectris X imaging filter (ThermoFisher), operated with a 20 eV slit size. Movie stacks were acquired using the EPU software (ThermoFisher) in super-resolution mode with a defocus range of −1.2 to −2.0 μm and a final calibrated pixel size of 0.932 Å. The total dose per EER (electron event representation) movie was 50 e-/Å².

## Cryo-EM data processing

For hZnT1 sample without $Zn^{2+}$ supplement, the motion of a total 4,879 movie stacks were corrected by MotionCor2 (Zheng et al, 2017) implemented in RELION (v3.1) (Zivanov et al, 2018). Exposure-weighted averages were first imported to cryoSPARC (v3.3.2) (Punjani et al, 2017), followed by the contrast transfer function estimated via CTFFIND4 (Rohou and Grigorieff, 2015). A total of 4,826,360 particles were kept after inspection of the blob-picking result and extracted with a box size of 220 pixels (binning 2×). A subset of particles was subject to 2D classification and the best representative averages were selected to reconstruct ab initio 3D model. Several rounds of 2D classification and heterogeneous refinement (3D classification) were then conducted on the whole particle set to remove contaminants or poor-quality particles. A

partition of 759,882 good particles were obtained and converted for RELION auto-refinement and Bayesian polishing. The shiny particles were then imported back to cryoSPARC for additional rounds of heterogeneous refinement, allowing 352,205 particles for subsequent homogenous refinement with C1 symmetry. Particles were further optimized with global and local CTF refinement. The final 3.48 Å map was reconstructed from local refinement with a mask of membrane micelle removed. Map resolution was estimated internally in cryoSPARC by gold-standard Fourier shell correlation using the 0.143 criterion. Details for data processing are described in Fig. EV1 and Table EV1.

For the 7158 EER movies of ZnT1-$Zn^{2+}$ sample collected on Falcon G4i detector, the EER movie of 1080 frames were fractionated into 40 subgroups and beam-induced motion was corrected with a MotionCor2-like algorithm implemented in RELION. Subsequent processing was conducted similarly as above. The final 2.65 Å map was reconstructed from local refinement with a mask of membrane micelle removed.

For the ZnT1 prepared at pH 6.0, a total of 2669 K3 movies were motion-corrected in RELION, imported to cryoSPARC for particle picking and classification. A set of 952,107 particles were then converted and conducted 3D classification in RELION (parameters: $K = 8$, $T = 10$). Three major maps with different conformations were identified and processed separately. Finally, maps for an inward-facing homodimer, an outward-facing homodimer and an inward-/outward-facing heterodimer were generated from 104,963 particles, 97,184 particles and 97,466 particles, respectively. Details for data processing are described in Fig. EV1 and Table EV1.

For ZnT1 supplemented with $Ca^{2+}$, total 4,476 K3 movies were collected and processed similarly as above. Final 3.52 Å map was reconstructed from 232,227 particles.

For ZnT3 prepared at pH 6.0, the 5,200 K3 movies were processed similarly as above. The final 3.14 Å map was reconstructed from 341,267 particles via local refinement with a mask of membrane micelle removed.

### Model building

Initial ZnT1 or ZnT3 models were retrieved from AphaFold database (Jumper et al, 2021). The predicted model was rigid-body docked into ZnT1 or ZnT3 cryo-EM density map in ChimeraX (v.1.6) (Goddard et al, 2018), followed by iterative manual adjustment in COOT (v.0.9.8) (Emsley et al, 2010) and real-space refinement in Phenix (v.1.19) (Adams et al, 2010). The model statistics were validated by Molprobity. Sidechains that do not have well-defined density were trimmed for deposition. The final refinement statistics are provided in Table EV1. Structural figures were prepared in ChimeraX or PyMOL (PyMOL Molecular Graphics SYtem, v.2.3.4, Schrödinger) (https://pymol.org/2/).

### Molecular dynamics simulations

We performed all-atom molecular dynamics (MD) simulations in explicit solvents for hZnT1 and hZnT3. The cryo-EM structures of the outward-facing $Zn^{2+}$-bound conformation of hZnT1 and the inward-open apo conformation of hZnT1 and hZnT3 were used as the starting coordinates for all simulations. For the inward-open apo conformation of hZnT1 or hZnT3, the initial model was generated by removing the $Zn^{2+}$ ions from inward-facing $Zn^{2+}$-

bound conformation. The transporter chain termini were capped with acetyl and methylamide. The missing loops between TM4-TM5 of hZnT1 and hZnT3 (residues 138–240 and 204–227, respectively) were omitted. PropKa was used to determine the dominant protonation state of all titratable residues at pH 7.4 (Olsson et al, 2011). The CHARMM-GUI Membrane builder module (Wu et al, 2014) was used to place each protein in a 1:1 POPC membrane patch with 20 Å of water above and below. The system was then solvated with TIP3P water molecules, and total 11 zinc ions were added manually in the proximity of the CTD for the simulation of inward-open apo conformation of hZnT3. For all simulations except the inward-open apo conformation of hZnT3 (zinc in bulk solvent), sodium and chloride ions were added at 150 mM to neutralize the system. For the simulation of inward-open apo conformation of hZnT3, 36 mM chloride ions were added to neutralize the system. The final MD simulation system of hZnT1 contained ~190 POPC lipids, and ~26,410 water molecules, with initial box dimensions of 90 Å × 90 Å × 153 Å. The final MD simulation system of hZnT3 had ~238 POPC lipids, and ~34,811 water molecules, with initial box dimensions of 115 Å × 115 Å × 130 Å. All of the simulations exploited the same force fields (FFs) for the lipid and the protein portions of the system, the Amber Lipid21 and the Amber ff14SB, respectively (Dickson et al, 2022; Maier et al, 2015). The non-bonded parameters for zinc (II) were taken in agreement with the use of TIP3P water model in combination with the classical 12-6 LJ non-bonded model (Li et al, 2013).

All MD simulations were conducted by Gromacs 2020.7 (Abraham et al, 2015). For each condition, three independent simulations were performed. All systems were energy minimized and equilibrated in six steps consisting of 2.5 ns long MD simulations, while slowly releasing the position restrain forces acting on the Cα atoms and zinc ions. Initial random velocities were assigned independently to each system. Production simulations without restrain were performed for 500 ns. The Verlet neighbor list was updated every 20 steps with a cut-off of 12 Å and a buffer tolerance of 0.005 kJ/mol/ps. Non-bonded van der Waals interactions were truncated between 10 and 12 Å using a force-based switching method. Long-range electrostatic interactions under periodic boundary conditions were evaluated using the smooth particle mesh Ewald method with a real-space cut-off of 12 Å (Steinbach and Brooks, 1994). Bonds to hydrogen atoms were constrained with the P-LINCS algorithm with an expansion order of four and one LINCS iteration (Hess, 2008). The constant temperature was maintained at 310 K using the v-rescale ($\tau = 0.1$ ps) thermostat (Bussi et al, 2007) by separately coupling solvent plus salt ions, membrane, and protein. Semi-isotropic pressure coupling was applied using the Parrinello-Rahman barostat (Parrinello and Rahman, 1981), using 1 bar and applying a coupling constant of 1 ps. Finally, a restrain-free production run was carried out with a time step of 2 fs (Chen et al, 2010; Gupta et al, 2014).

## Data availability

The coordinates for ZnT1 models have been deposited in the PDB under accession code 8XM6 ($Zn^{2+}$-free OF/OF), 8XMA ($Zn^{2+}$-bound OF/OF), 8XMF ($Zn^{2+}$-bound IF/IF), and 8XMJ ($Zn^{2+}$-bound

IF/OF). The coordinate for ZnT3 has been deposited under code 8XN1. The cryo-EM density maps have been deposited in the Electron Microscopy Data Bank with accession code EMD-38465, EMD-38469, EMD-38475, EMD-38479, EMD-38474, and EMD-38494.

The source data of this paper are collected in the following database record: biostudies:S-SCDT-10_1038-S44319-024-00287-3.

## Peer review information

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

## Acknowledgements

The authors thank the Center of Cryo-Electron Microscopy, as well as confocal core facility and multifunctional platform for biomedical imaging analysis (SP8 LSCM), at Core Facility of Shanghai Medical College, Fudan University for technical support and assistance. This work was supported by the National Natural Science Foundation of China (32171194 and 32371256 to QQ), the National Key R&D Program of China (2023YFA0915000 to QQ), and the China Postdoctoral Science Foundation (2022M720805 to Z Zhou).

## Author contributions

**Yonghui Long**: Data curation; Formal analysis; Investigation; Methodology; Writing—original draft; Writing—review and editing. **Zhini Zhu**: Data curation; Formal analysis; Investigation; Methodology; Writing—original draft. **Zixuan Zhou**: Data curation; Software; Formal analysis; Validation; Investigation. **Chuanhui Yang**: Data curation; Software; Formal analysis; Investigation; Methodology; Writing—original draft; Writing—review and editing. **Yulin Chao**:

Data curation; Formal analysis; Investigation; Methodology. **Yuwei Wang**: Investigation; Methodology. **Qingtong Zhou**: Software; Methodology; Writing—review and editing. **Ming-Wei Wang**: Software; Methodology; Writing—review and editing. **Qianhui Qu**: Conceptualization; Formal analysis; Supervision; Funding acquisition; Validation; Investigation; Visualization; Methodology; Writing—original draft; Project administration; Writing—review and editing.

Source data underlying figure panels in this paper may have individual authorship assigned. Where available, figure panel/source data authorship is listed in the following database record: biostudies:S-SCDT-10_1038-S44319-024-00287-3.

## Disclosure and competing interests statement

The authors declare no competing interests.

# Expanded View Figures

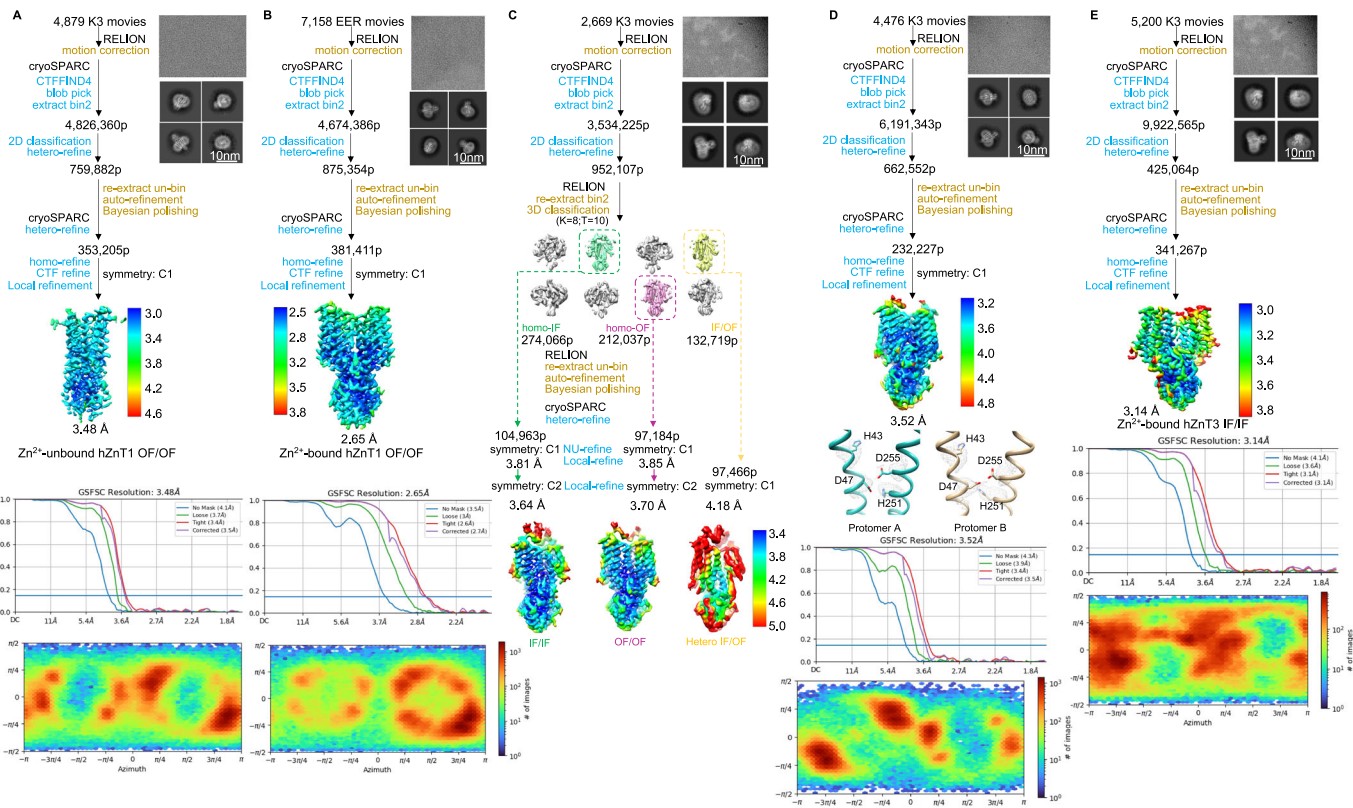

**Figure EV1. Cryo-EM processing of hZnT1 and hZnT3 samples at different conditions.**

(A) Data processing flowchart of hZnT1 in the absence of any ligand, at pH 7.5. Representative raw micrographs and 2D classifications are shown. (B) Data analysis flowchart of hZnT1 in the presence of 1 mM $Zn^{2+}$, at pH 7.5. (C) Processing flowchart of hZnT1 in the presence of 1 mM $Zn^{2+}$, at pH 6.0. (D) Data processing workflow of hZnT1 with 1 mM $Ca^{2+}$ supplement, at pH 7.5. (E) Data processing workflow of hZnT3 with 1 mM $Zn^{2+}$ supplement, at pH 6.0.

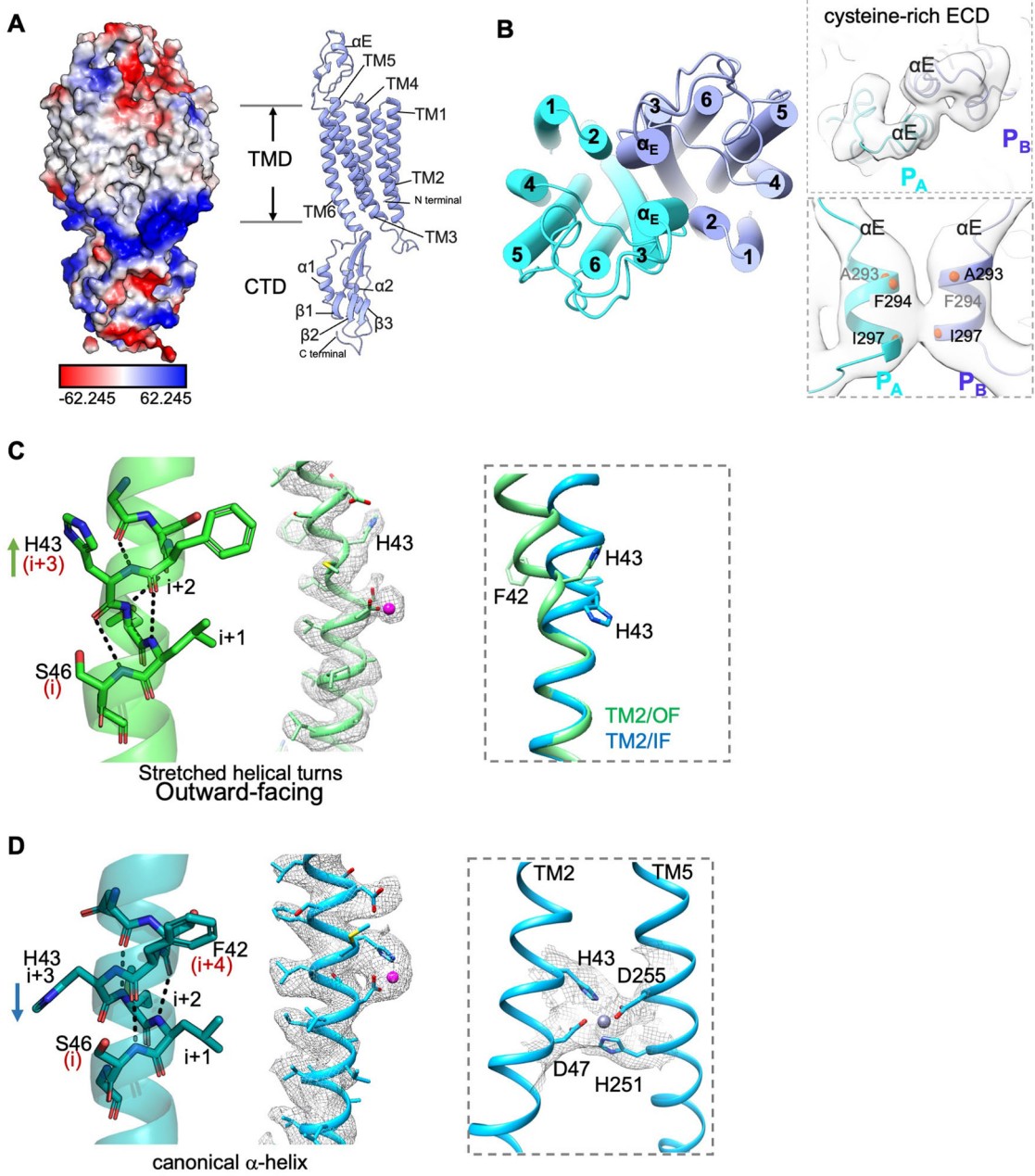

**Figure EV2. Unique structural features of hZnT1.**

(A) Surface of hZnT1 dimer is rendered in electrostatic potential. Cartoon view of one protomer, with segments labeled accordingly (right). (B) The extracellular cysteine-rich loops fit in unsharpened map (left), with two short helices forming the interface. (C) The TM2 segment around His43 residues in the $Zn^{2+}$-bound outward-facing dimer adopted a stretched conformation, with His43 sidechain pointing toward extracellular side. A comparison with canonical TM2 helix was shown on right. (D) The TM2 segment around His43 residue in the $Zn^{2+}$-bound inward-facing dimer adopted a canonical a-helix with His43 sidechain pointing towards the coordinated $Zn^{2+}$. The main-chain hydrogen bonds are shown as black dashed lines. The tetrahedral coordination network was highlighted on right with mesh density shown for the four key residues.

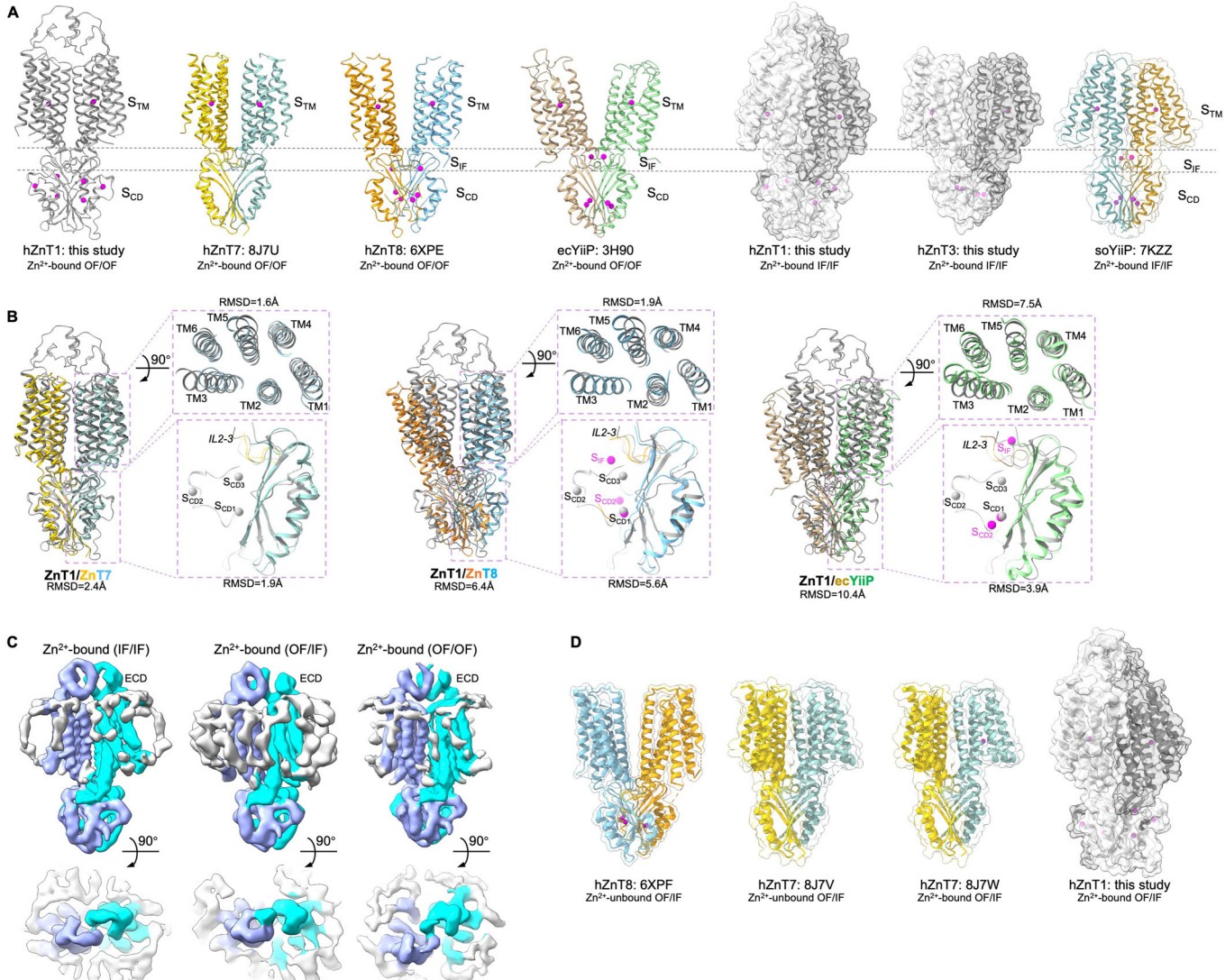

**Figure EV3. Structural comparison of human ZnTs and bacterial YiiP proteins.**

(**A**) Comparison of the $Zn^{2+}$-bound homologous outward-facing structures of hZnT1 (this study), hZnT7 (PDB: 8J7U), hZnT8 (PDB: 6XPE) and *E. coli* ecYiiP (PDB: 3H90), and inward-facing homodimers hZnT1 (this study), hZnT3 (this study), and soYiiP (PDB: 7KZZ). $Zn^{2+}$ binding sites are labeled in accordance. hZnT1 and hZnT3 in this study were prepared similarly in LMNG/CHS detergent micelle and determined by single-particle cryo-EM. hZnT7 was prepared in GDN detergent micelle at pH 7.5 and determined by single-particle cryo-EM with a high-affinity Fab. hZnT8 was prepared in digitonin detergent micelle at pH 7.4 and determined by single-particle cryo-EM. EcYiiP was prepared in n-undecyl-β-D-maltoside at pH 7.0 and determined by X-ray crystallography. SoYiiP was prepared in n-decyl-β-D-maltoside at pH 7.5 and determined by crystallography cryo-EM. (**B**) Superimpose of hZnT1 with ZnT7, ZnT8 and ecYiiP. The TMD and CTD regions are highlighted on right, with RMSD values shown in accordance. (**C**) Comparison of the three $Zn^{2+}$-bound hZnT1 maps obtained at pH 6.0. EM densities are shown at the same contour level ($\sigma = 6.0$). The IF/IF homodimer presents slightly better quality of dimeric ECD region compared to OF/IF heterodimer and OF/OF homodimer. (**D**) Overall structures of the $Zn^{2+}$-bound OF/IF hZnT1 heterodimer (this study), $Zn^{2+}$-unbound OF/IF hZnT8 (PDB: 6XPF) and hZnT7 (PDB: 8J7V), as well as $Zn^{2+}$-bound OF/IF hZnT7 (PDB: 8J7W).

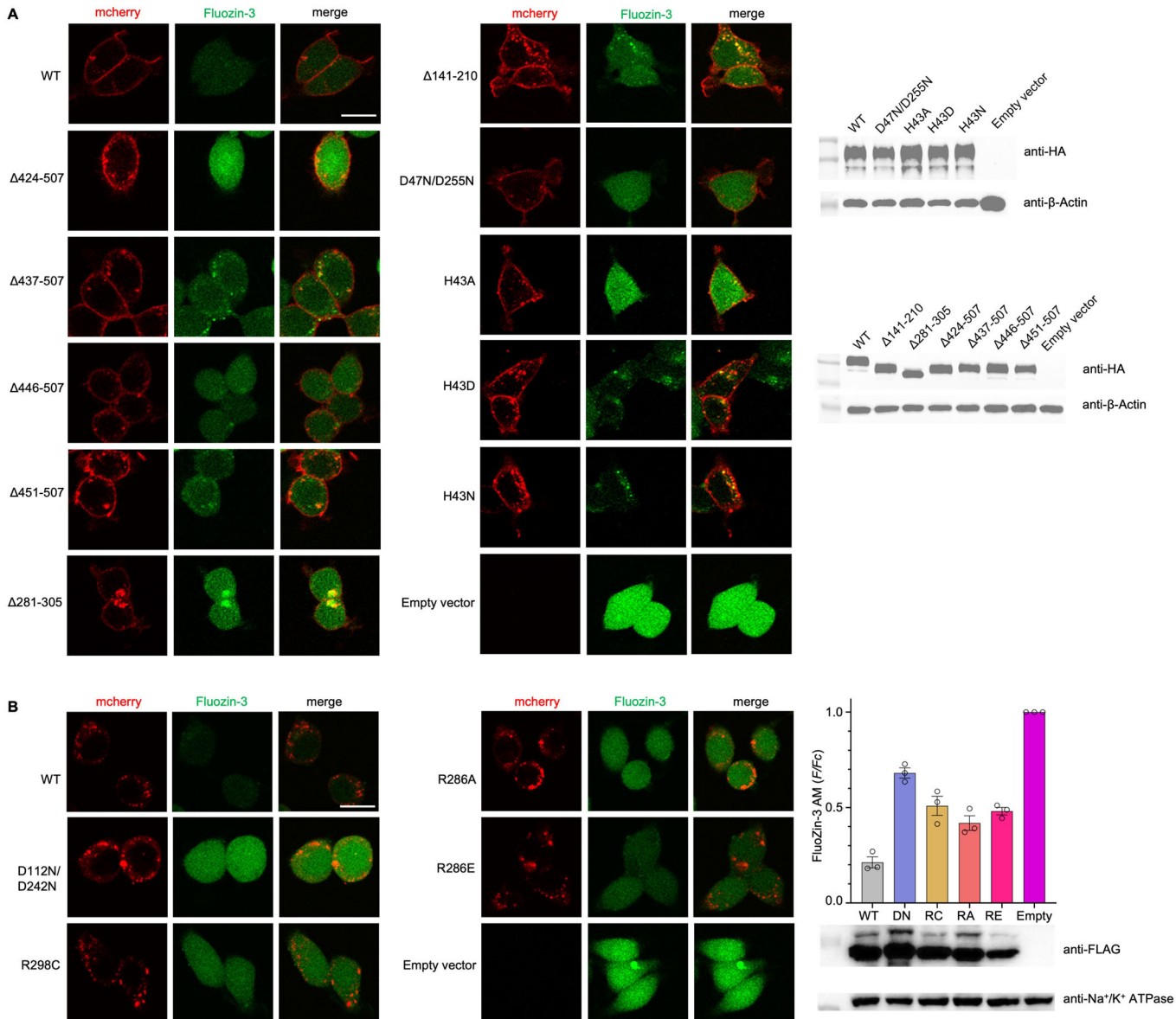

**Figure EV4. Representative fluorescent images of cellular efflux activity of hZnT1 and hZnT3 WT and mutants.**

(A) ZnT1 wild-type (WT) or mutants are fused with mCherry for visualization in stably transfected HEK293T cells. Cells with similar red fluorescence, which indicates protein expression levels, are selected for data analysis. Intracellular green fluorescence intensity was measured and analyzed in ImageJ. Cells transfected with an empty vector alone are used as control. Protein expression levels of WT ZnT1 and mutants were probed with anti-HA antibody, using internal β-actin as loading control. Scale bar size, 20 μm. (B) ZnT3 constructs are tagged with mCherry for visualization in stably transfected HEK293T cells. $n > 181$ total cells of intracellular FluoZin-3 AM fluorescence were analyzed for each group. Error bars indicate means ± SEM, $N = 3$ independent experiments. Total protein expression level of WT ZnT3 and mutants were measured using anti-Flag antibody, with anti-Sodium Potassium ATPase ($Na^+/K^+$ ATPase) as loading control. Scale bar size, 20 μm. Source data are available online for this figure.

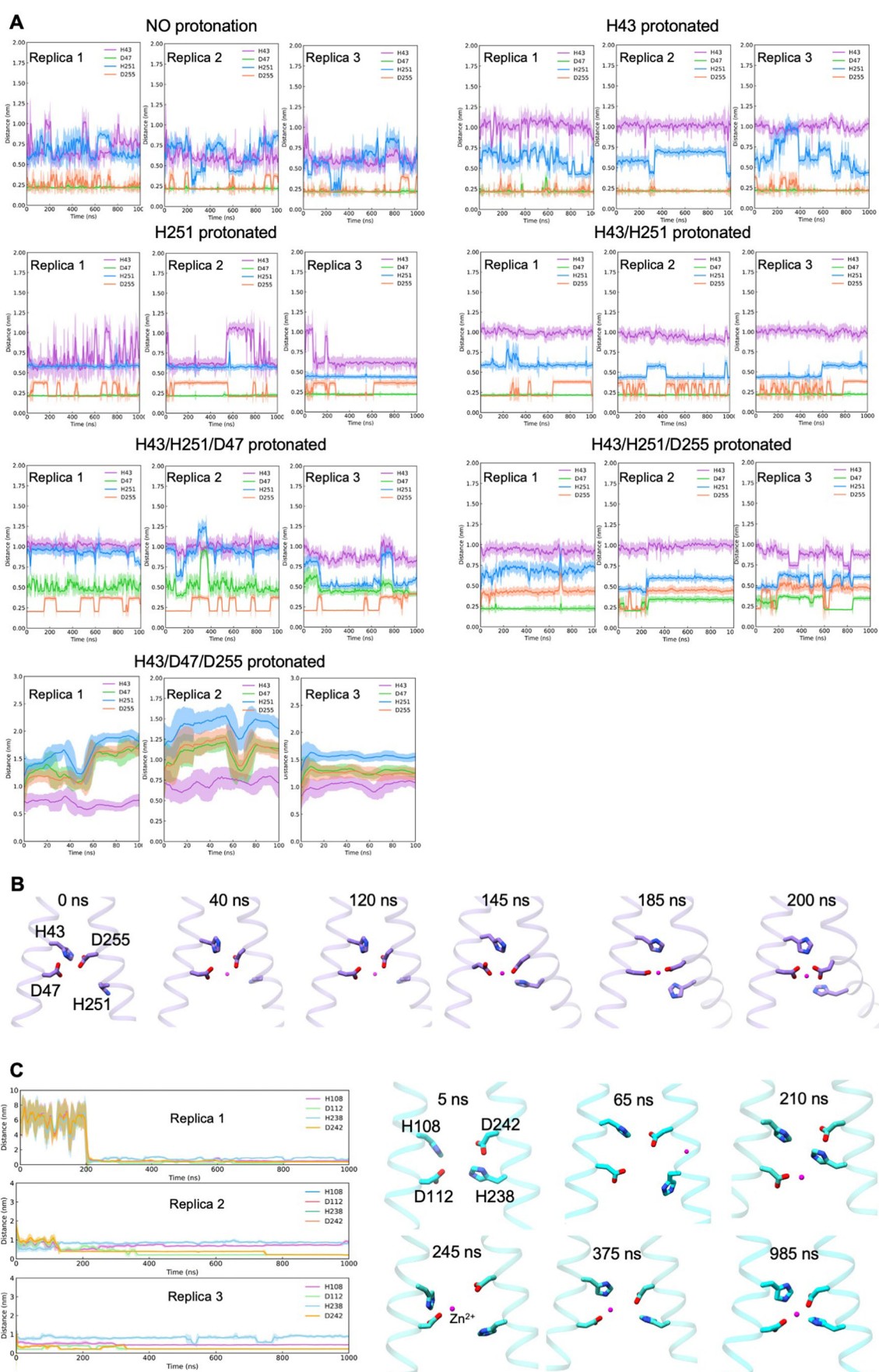

◀  **Figure EV5.  MD simulation analysis of hZnT1 and hZnT3.**

(A) The distances between center of mass of $Zn^{2+}$ ions and sidechains of H43, D47, H251 and D255 on hZnT1, with protonation sates indicated, in three replicas. (B) Representative views of $Zn^{2+}$ ion enters and binds the tetrahedral coordination network of inward-facing hZnT1 at time points indicated. (C) The distances between center of mass of $Zn^{2+}$ ions and sidechains of H108, D112, H238 and D242 on inward-facing hZnT3. Representative views of $Zn^{2+}$ ion enters and binds the tetrahedral coordination network are shown on the right.

