## [Peer Review File · EMBO Reports]

Structural insights into the human zinc transporter ZnT1 mediated Zn²⁺ efflux

Qianhui Qu, Yonghui Long, Zhini Zhu, Zixuan Zhou, Chuanhui Yang, Yulin Chao, Yuwei Wang, Qingtong Zhou, and Ming-Wei Wang

Corresponding author(s): Qianhui Qu (qqh@fudan.edu.cn)

Review Timeline:

Submission Date:	1st Feb 24
Editorial Decision:	4th Mar 24
Revision Received:	1st Jun 24
Editorial Decision:	15th Jul 24
Revision Received:	26th Jul 24
Editorial Decision:	4th Sep 24
Revision Received:	6th Sep 24
Accepted:	18th Sep 24

Transaction Report:

Dear Dr. Qu

Thank you for the submission of your research manuscript to our journal. We have now received the full set of referee reports that is copied below.

As you will see, the referees raise very significant concerns regarding your study. Referee 1 considers the structural part insufficient, with apparently good maps but incorrect tracing. Referee 2 agreed with these concerns upon further discussion of the reports. Despite these limitations, both referees would support publication in EMBO Reports, under the condition that the structural part is redone, the wrong tracing corrected and the maps rebuilt. In addition, the efflux experiments need to be redone and the conditions must be clearly outlined. Also, the MD simulations need to be expanded.

From the referee comments it is clear that, as it stands, the technical quality of the study is low/unacceptable and publication of the manuscript in our journal can therefore not be considered at this stage. On the other hand, given the potential interest of your findings, I would like to give you the opportunity to address the concerns and would be willing to consider a revised manuscript with the understanding that the referee concerns must be fully addressed and their suggestions (as detailed above and in their reports) taken on board. All concerns listed above and in the referee reports are important and need to be addressed.

Should you decide to embark on such a revision, acceptance of the manuscript will depend on a positive outcome of a second round of review and I should also remind you that it is EMBO reports policy to allow a single round of revision only and that, therefore, acceptance or rejection of the manuscript will depend on the completeness of your responses included in the next, final version of the manuscript.

We realize that it is difficult to revise to a specific deadline. In the interest of protecting the conceptual advance provided by the work, we recommend a revision within 3 months (June 4). Please discuss the revision progress ahead of this time with the editor if you require more time to complete the revisions.

I am also happy to discuss the revision further via e-mail or a video call, if you wish.

*******IMPORTANT NOTE:**

We perform an initial quality control of all revised manuscripts before re-review. Your manuscript will FAIL this control and the handling will be delayed IN CASE the following APPLIES:

- 1) A data availability section providing access to data deposited in public databases is missing. If you have not deposited any data, please add a sentence to the data availability section that explains that.
- 2) Your manuscript contains statistics and error bars based on $n=2$. Please use scatter blots in these cases. No statistics should be calculated if $n=2$.

When submitting your revised manuscript, please carefully review the instructions that follow below. Failure to include requested items will delay the evaluation of your revision. *****

2) individual production quality figure files as .eps, .tif, .jpg (one file per figure).

Please download our Figure Preparation Guidelines (figure preparation pdf) from our Author Guidelines pages <https://www.embopress.org/page/journal/14693178/authorguide> for more info on how to prepare your figures.

4) a complete author checklist, which you can download from our author guidelines (<https://www.embopress.org/page/journal/14693178/authorguide>). Please insert information in the checklist that is also reflected in the manuscript. The completed author checklist will also be part of the RPF.

5) Please note that all corresponding authors are required to supply an ORCID ID for their name upon submission of a revised manuscript (<https://orcid.org/>). Please find instructions on how to link your ORCID ID to your account in our manuscript tracking system in our Author guidelines (<https://www.embopress.org/page/journal/14693178/authorguide#authorshipguidelines>)

6) We replaced Supplementary Information with Expanded View (EV) Figures and Tables that are collapsible/expandable online. A maximum of 5 EV Figures can be typeset. EV Figures should be cited as 'Figure EV1, Figure EV2' etc... in the text and their respective legends should be included in the main text after the legends of regular figures.

7) Before submitting your revision, primary datasets (and computer code, where appropriate) produced in this study need to be deposited in an appropriate public database (see <https://www.embopress.org/page/journal/14693178/authorguide#dataavailability>).

The accession numbers and database should be listed in a formal "Data Availability " section (placed after Materials & Method) that follows the model below (see also <https://www.embopress.org/page/journal/14693178/authorguide#dataavailability>). Please note that the Data Availability Section is restricted to new primary data that are part of this study.

Data availability

Additional information on source data and instruction on how to label the files are available <https://www.embopress.org/page/journal/14693178/authorguide#sourcedata>.

10) Figure legends and data quantification:
The following points must be specified in each figure legend:

- the name of the statistical test used to generate error bars and P values,
- the number (n) of independent experiments (please specify technical or biological replicates) underlying each data point,
- the nature of the bars and error bars (s.d., s.e.m.)

- If the data are obtained from n {less than or equal to} 5, show the individual data points in addition to the SD or SEM.
- If the data are obtained from n {less than or equal to} 2, use scatter blots showing the individual data points.

Discussion of statistical methodology can be reported in the materials and methods section, but figure legends should contain a

basic description of n, P and the test applied.

12) All Materials and Methods need to be described in the main text. We would encourage you to use 'Structured Methods', our new Materials and Methods format. According to this format, the Materials and Methods section should include a Reagents and Tools Table (listing key reagents, experimental models, software and relevant equipment and including their sources and relevant identifiers) followed by a Methods and Protocols section in which we encourage the authors to describe their methods using a step-by-step protocol format with bullet points, to facilitate the adoption of the methodologies across labs. More information on how to adhere to this format as well as downloadable templates (.doc or .xls) for the Reagents and Tools Table can be found in our author guidelines: <

<https://www.embopress.org/page/journal/14693178/authorguide#manuscriptpreparation>>. An example of a Method paper with Structured Methods can be found here: <<https://www.embopress.org/doi/10.15252/msb.20178071>>.

11) Our journal encourages inclusion of *data citations in the reference list* to directly cite datasets that were re-used and obtained from public databases. Data citations in the article text are distinct from normal bibliographical citations and should directly link to the database records from which the data can be accessed. In the main text, data citations are formatted as follows: "Data ref: Smith et al, 2001" or "Data ref: NCBI Sequence Read Archive PRJNA342805, 2017". In the Reference list, data citations must be labeled with "[DATASET]". A data reference must provide the database name, accession number/identifiers and a resolvable link to the landing page from which the data can be accessed at the end of the reference. Further instructions are available at <<https://www.embopress.org/page/journal/14693178/authorguide#referencesformat>>.

12) As part of the EMBO publication's Transparent Editorial Process, EMBO Reports publishes online a Review Process File to accompany accepted manuscripts. This File will be published in conjunction with your paper and will include the referee reports, your point-by-point response and all pertinent correspondence relating to the manuscript.

Yours sincerely,

Referee #1:

The paper by Long et al., reports the structure of hZnT1 solved by cryo-EM in various states of metal binding and includes the structure of hZnT3. The authors sought to understand the transport mechanism and compare it to other ZnTs that have been determined. To shed some light on the importance of specific locations, such as the transport site and other relevant domains in the protein, Zn transport was reported to compare WT and mutants. ZnT1 is an important protein that is more complicated than the other ZnTs, and several researchers were looking for its structure. The authors were able to find relevant structures in several transport stages and suggested a mechanism for its structure.

The structural part suffers from low craftsmanship. The PDB files contain many errors; some parts of the structure do not fit the map, and residues have the wrong rotamers or are missing density, including wrong map tracing in some sections. Some domains are missing from the map but were added without relevant density. I will only give some examples out of many. Using common tools in Coot (used by the authors), such as real space refine zone and sphere refine, I got a better structure for the

map.

Examples:

in 8XMJ (a low-resolution map)

- 1) missing map to 422-427
- 2) additional density after 136

in 8XMA high res map

- 1) missing map to 422-427
- 2) wrong tracing of 438-445 with forcing a cys for Zn binding with density clearly going to other directions
- 3) The GLN side chain occupies the place of the main chain, and there is a clear continuation of the helix without its tracing.
- 4) residues 34-44 are clearly out of the map.
- 5) No map for 275-308

in 8XM6

- 1) 41-45 The main chain and side chains are not sitting well on the map. The difference between chains A and B is lower than the error based on the map resolution. By fixing the problems the generated structure is different from both. This is critical as the discussion of moving between helix 3-10 to canonical helix is not real and is not supported by the map. It reduces the paper quality and again presents that the structural analysis done is wrong and the whole structure should be fixed.

In addition to the structural problems, there are numerous issues in the methodology the authors used to assess the activity of the transporters and their mutants.

First, for loading FluZin-3, the authors incubated the cells for 50 minutes at 37C. It has been known that temperature affects the cellular localization of ion-sensitive dyes (Roe et. al Cell Calcium 1990, Trolinger et. al Biophysical J. 2000). Consequently, the fluorescent observed would not reflect cytosolic Zn²⁺ but rather a combined outcome of Zn²⁺ in the cytosol plus its concentrations due the contribution of the dye localized to different intracellular compartments.

The method used for evaluating the activity of the different transporters is not clear.

Over what period was efflux measured?

Was it assessed during the initial rate of efflux?

These points should be stated in the text and the test should be shown in the supplement.

Quantitative evaluation of efflux cannot be accurately evaluated if it is not based on measurements made when the transport process is at its initial rate for a period. During this time, conditions are defined before gradients and driving forces are altered. Furthermore, I wonder what was measured in the efflux experiment as there was no calcium ion in the buffer, which is essential for the ZnT1 to extrude Zn²⁺ out of the cell (Shusterman et. al. Metallomics, 2014). Moreover, the role of Ca is so important that it was even speculated to be a Zn²⁺/Ca²⁺ exchanger (Gottesman et. al., Cell Calcium., 2022).

Some minor points:

- 1) In the molecular dynamics simulation, the authors removed the histidine-rich loop, and it is unclear if the created termini stayed open. If yes, it has a significant influence on the simulation, and it needs to be addressed.
- 2) There is no discussion on the histidine-rich loop, which is one of the main differences between ZnT1 and other ZnTs.
- 3) Figure EV4 C and D, additional panels should be added with the map around these helices. Since the transition between Helix 3-10 and canonical helix are one of the main points, it is not a match to the map, and what is the result of rotating the helix on the 3-10 forms other residues and other interactions.

Referee #2:

This paper represents a significant advancement in the zinc transporter research field by elucidating multiple states of zinc transporter 1 (ZnT1) through cryo-electron microscopy, capturing snapshots with zinc ions. Furthermore, molecular dynamics (MD) simulations analysis demonstrated that the protonation on His residues influences their interaction with the Zn ion. However, the paper has some weak points.

The mechanism by which proton-driven significant structural changes are elicited awaits further analysis. While mutational functional analysis in vivo was performed, definitive information on the molecular mechanisms was not obtained. Although lipid-related function is discussed in the later part of the paper, the assumption may want to be elucidated by measuring Zn transporter activity in vitro.

While MD simulations focus on Zn binding sites, if structural changes in the transmembrane region were observed, it could further support the proposed model. At least, more elaboration on MD simulations about alpha helices movements would be beneficial.

Addressing these points would improve the paper, but I encourage an early publication, given the impact of reporting multiple structures of the same protein. It might be appropriate to put unresolved issues in future studies.

Minor Comments

Lines 49 and 58: use the same abbreviation: SLC30/ZnT or ZnT/SLC30.

In the MD simulations, please provide an additional explanation regarding alpha helices changes.

In material and method,

Please provide the necessary information for constructing the proteins, including linkers and tags, which is essential for ensuring the reproducibility of the experiments. Additionally, compiling a plasmid list would be helpful.

HEPES should be HEPES-Na or HEPES-K

Please provide the name of the maker of the DYKDDDDK(FLAG) affinity resin.

Lines 493 and 519: ZnSO₄: 4 should be subscript.

Figs. 1B and EV4: please indicate the range of electrostatic potential.

Fig. EV4C and B bottom right, please include an EM map.

Figs. EV5 and EV6: please include sequence IDs for easy access.

Fig. EV7: for clarity and comparison, please specify the method of structure acquisition and the solubilization state (or embedded in nanodiscs). Additionally, please include RMSD and the regions that were superimposed.

Fig. EV8: since there is no description of the Western blotting data, please include it.

Fig. EV13: if possible, provide the accumulation data similar to EV8.

Referee #1:

The paper by Long et al., reports the structure of hZnT1 solved by cryo-EM in various states of metal binding and includes the structure of hZnT3. The authors sought to understand the transport mechanism and compare it to other ZnTs that have been determined. To shed some light on the importance of specific locations, such as the transport site and other relevant domains in the protein, Zn transport was reported to compare WT and mutants. ZnT1 is an important protein that is more complicated than the other ZnTs, and several researchers were looking for its structure. The authors were able to find relevant structures in several transport stages and suggested a mechanism for its structure.

We thank the Reviewer for the insightful comments and strong support of our work.

The structural part suffers from low craftsmanship. The PDB files contain many errors; some parts of the structure do not fit the map, and residues have the wrong rotamers or are missing density, including wrong map tracing in some sections. Some domains are missing from the map but were added without relevant density. I will only give some examples out of many. Using common tools in Coot (used by the authors), such as real space refine zone and sphere refine, I got a better structure for the map.

The constructive comments are appreciated. We apologize for any inadvertent inaccuracies and have carefully rebuilt the structural modelling as indicated by the Reviewer. The few periphery regions that are not well resolved in the sharpened map, wherein their initial model tracing was guided by the unsharpened map and AlphaFold2 prediction, are now removed from the formally deposited model. Despite the modification, we feel that the scientific conclusions presented in the initial submitted version are not significantly affected by the rebuilt models.

Examples:

in 8XMJ (a low-resolution map)

1) missing map to 422-427

Points are taken. The peripheral region 422-427 on CTD domain, which was previously modelled into unsharpened map based on AlphaFold2 prediction, is now removed in the new model.

2) additional density after 136

We thank the Reviewer for catching the short additional density after 136 of the peripheral TM4 helix. We now rebuild two more residues into the weak density with sidechains trimmed in new models.

in 8XMA high res map

1) missing map to 422-427

As stated above, the same region 422-427 is now deleted in the new model.

2) wrong tracing of 438-445 with forcing a cys for Zn binding with density clearly going to other directions

Thanks for pointing this out. It is now corrected as seen in the following Figure x1.

Figure x1. Rebuilt region (438-445) into the density.

3) The GLN side chain occupies the place of the main chain, and there is a clear continuation of the helix without its tracing.

Thanks for pointing this out. The error is now corrected as shown in above Figure x1.

4) residues 34-44 are clearly out of the map.

Thanks for catching this outlier in the Protomer A model of ZnT1 dimer structure. We rebuilt the regions as follows:

Figure x2. The regions (34-44) are rebuilt into the density map (contour level $\sigma=4.5$).

5) No map for 275-308

ZnT1 has a distinct extracellular region (275-308 residues) that is rich in cysteines. This region was initially modelled using AlphaFold2 prediction (Figure x3A), then manually adjusted and refined against the unsharpened outward-facing map (Figures x3B and x3C). Due to the lack of high-quality map density, we merely traced the main-chain for this region in previous models, and did not describe any specific interactions in the manuscript. To minimize misunderstanding, we have deleted regions without clear density for deposition.

Figure x3. The extracellular cysteine-rich region (275-308), which was based on AlphaFold2 prediction (A), and modelled (B) into unsharpened map (C). Density map was shown as semi-transparent surface.

in 8XM6

1) 41-45 The main chain and side chains are not sitting well on the map. The difference between chains A and B is lower than the error based on the map resolution. By fixing the problems the generated structure is different from both. This is critical as the discussion of moving between helix 3-10 to canonical helix is not real and is

not supported by the map. It reduces the paper quality and again presents that the structural analysis done is wrong and the whole structure should be fixed.

Thanks for raising this concern. In the 2.6-Å Zn^{2+} -bound outward-facing ZnT1 map, Protomer A's TM2 segment (residues 41-45) may have a mixed conformation (i.e., 3_{10} helix and canonical helix), while Protomer B's TM2 (41-45) fits well in the density and folds primarily as a 3_{10} helix. To resolve potential mixed conformations in Protomer A, we applied different classification strategies, including focused 3D classification without particle alignment (mask on Protomer A TMs region in RELION), 3D classification with different reference maps (in cryoSPARC), and 3D variability analysis (in cryoSPARC), to the 2.6-Å particles set (345397 particles).

After many rounds of skip-alignment 3D classification and refinement (Figure x4), we were able to improve the local quality of Protomer A TM2 segment, resulting in two substates in terms of His43 orientation (substate 1, His43 points toward extracellular side, 2.96-Å; substate 2, His43 points toward intracellular side and interacts with Zn^{2+} ligand, 2.99-Å). In the His43 up substate, the TM2 (41-45) segment remains in 3_{10} helix (Figure x5A). While in the His43 down substate, this segment resembles a canonical helix (Figure x5B). Despite our significant categorization efforts, we admit that the model's fit into the density is still suboptimal, presumably because of the residual mixed conformation particles in the map.

Of note, the TM2 segment (41-45) in the 3.6-Å Zn^{2+} -bound inward-facing model exhibits a canonical helix shape (Figure x5C). When ZnT1 transitions from an inward-facing to an outward-facing state, the TM2 segment may move from the canonical helix to helix 3_{10} , releasing Zn^{2+} into the extracellular space. Thus, we believe that the newly generated models have no substantial impact on our explanation of these local conformational changes in the initial manuscript.

Figure x4. Focused 3D classification on Protomer A TM region generates two substates for Zn^{2+} -bound outward-facing ZnT1.

Figure x5. TM2 conformational difference in Zn^{2+} -bound outward-facing (A,B) and inward-facing ZnT1 (C) models.

In addition to the structural problems, there are numerous issues in the methodology the authors used to assess the activity of the transporters and their mutants.

We thank the Reviewer for the following valuable suggestions to improve our manuscript.

First, for loading FluoZin-3, the authors incubated the cells for 50 minutes at 37°C. It has been known that temperature affects the cellular localization of ion-sensitive dyes (Roe et. al Cell Calcium 1990, Trolinger et. al Biophysical J. 2000). Consequently, the fluorescent observed would not reflect cytosolic Zn²⁺ but rather a combined outcome of Zn²⁺ in the cytosol plus its concentrations due the contribution of the dye localized to different intracellular compartments.

Figure x6. Retentive intracellular FluoZin-3 fluorescence intensity were measured for WT ZnT1 or control cells at different experimental temperature for varied periods of time. For each group, total 250~300 cells were recorded by confocal microscope, and the intensity values were calculated in ImageJ.

We agree with the Reviewer that the cellular localization of membrane-permeable FluoZin-3 AM indicator would be affected by temperature and incubation time. Therefore, we first measured the retentive intracellular fluorescence in wildtype ZnT1-stably transfected cells or control cells under different temperatures (37 °C or room temperature), for 20min or 50min. Total 250~300 cells were recorded by a Leica SP8 LSCM+ laser scanning confocal microscope for each group. As shown in Figure x6, the temperature and loading period do have an effect on the recorded fluorescent signal. However, all groups conclude that exogenously expressed WT ZnT1 cells have higher Zn²⁺ efflux capacity than control cells. In particular, the experimental setting of treating cells at 37 °C for 50min resulted in the most significant difference. It is possible that the fluorescent signal recorded under these conditions is a combined outcome as pointed out by the Reviewer. However, since all the WT and mutant ZnT1 cells were treated in the same way, we believe that the retentive intracellular fluorescent signal can roughly indicate the Zn²⁺ efflux capacity of ZnT1 variants. We then measured the intracellular FluoZin-3 AM retention signal for different ZnT1 mutants at 22 °C for 50min (Figure x7A), and at 37 °C for 50min

(Figure x7B). The overall trend of fluorescent retention is consistent among ZnT1 variants between these two experimental settings. We therefore chose this retentive fluorescent signal method to assess the Zn²⁺ efflux capacity of WT ZnT1 and related mutants.

Figure x7. Retentive intracellular FluoZin-3 fluorescence measurement for WT ZnT1 and mutants at different temperatures (A, 22°C; B, 37°C). A, ~100 cells were measured for each group, in one experiment. B, at least 80 cells were measured for each experiment and were repeated three times biologically. The fluorescence signal (*F*) is normalized to that of the empty vector (*F_c*). Significance was analyzed by one-way ANOVA; *****p*<0.0001.

The method used for evaluating the activity of the different transporters is not clear. Over what period was efflux measured?

Was it assessed during the initial rate of efflux?

These points should be stated in the text and the test should be shown in the supplement.

Quantitative evaluation of efflux cannot be accurately evaluated if it is not based on measurements made when the transport process is at its initial rate for a period. During this time, conditions are defined before gradients and driving forces are altered.

We apologize for any unclear description about the efflux method. The WT or mutants stably transfected HEK293T cells were seeded to the transparent glass-bottom petri dishes, which can be imaged directly by confocal microscope. The efflux experiments were conducted in DMEM medium (Gibco, cat. C11995500BT) which contains 1.8 mM Ca²⁺ according to the manufacture document, at 37 °C for 50min. The cells were then washed three times quickly and recorded on a Leica SP8 LSCM+ laser scanning confocal microscope. The intracellular FluoZin-3 AM

fluorescent signals of 100~200 cells were calculated for each group. As stated above, we also assessed the potential effect of temperature and time, and observed a consistent efflux outcome among different test groups. We agree with the Reviewer that quantitative evaluation of efflux cannot be accurately evaluated if it is not based on measurements at its initial efflux rate. We admit that the accumulative/retentive measurement of FluoZin-3 AM signal over 50min is a suboptimal strategy. However, we believe that the efflux capacity of WT ZnT1 or mutants can be analysed qualitatively by this strategy, given the consistent trend measured under conditions of different temperatures and incubation time. We now included the test data (Figures x6 and x7) in the supplement as Appendix Figure S5, and make changes accordingly in the text.

Furthermore, I wonder what was measured in the efflux experiment as there was no calcium ion in the buffer, which is essential for the ZnT1 to extrude Zn²⁺ out of the cell (Shusterman et. al. *Metallomics*, 2014). Moreover, the role of Ca is so important that it was even speculated to be a Zn²⁺/Ca²⁺ exchanger (Gottesman et. al., *Cell Calcium*, 2022).

Thanks for raising this point. The cellular efflux experiments were conducted in the DMEM basic medium (Gibco, cat. C11995500BT) which contains 1.8 mM Ca²⁺ in the formula, according to the manufacture document. Therefore it suffices the requirement for Ca²⁺. The buffer (20 mM HEPES pH 7.4, 125 mM KCl, 5 mM NaCl, 10 mM Glucose, and 10 mM phenanthroline) mentioned in the Method was used to wash cells for subsequent microscopy imaging.

We were also encouraged by the two reports on the role of Ca²⁺ highlighted by the Reviewer, to investigate the molecular mechanism of Zn²⁺/Ca²⁺ exchange. To that end, we performed cryo-EM analysis on the ZnT1/Ca²⁺ sample, which was prepared similarly to the Zn²⁺-bound ZnT1 samples. Unfortunately, no discernible extra density corresponding to Ca²⁺ could be found in the TMD region of a 3.52-Å map (Figure x8). As a result, our pursuit of Zn²⁺/Ca²⁺ exchange mechanism was discontinued. Meanwhile, we captured ZnT1's conformational dynamics (OF/OF, OF/IF and IF/IF structures) only in the presence of both low pH (H⁺) and Zn²⁺, which may support the H⁺-driven Zn²⁺ transport event observed by Shusterman and colleagues (Shusterman et al., *Metallomics*, 2014).

Of note, during the revision of our manuscript, Sun and co-workers determined a cryo-EM structure of ZnT1 in the Apo state at 3.4-Å resolution, and observed the Ca²⁺ dependence on ZnT1-mediated Zn²⁺ transport in both cellular assays and *in vitro* proteoliposomes (Sun et al., *Science Advance*, 2024). However, the lack of a Ca²⁺-bound ZnT1 structure or evidence of direct Ca²⁺/ZnT1 binding limits our understanding of the precise role Ca²⁺ plays in ZnT1-mediated Zn²⁺ efflux. It is also unclear if Ca²⁺ can bind to ZnT1 in the same way as the cognate substrate Zn²⁺ does.

In line with the comments by the Reviewer that ZnT1 is more complicated than the other ZnTs, recent studies also showed that ZnT1 can localize to intracellular compartments (Abdo et al., 2021), such as mitochondria in rat hepatocytes (Sun et al., 2015) and endosome in human macrophages (Yang et al., 2023). Given the pH difference between these compartments and the cytosol, we speculate an important role for proton driving ZnT1-mediated Zn^{2+} transport across these intracellular membranes.

Figure x8. Cryo-EM processing of ZnT1 in the presence of Ca^{2+} .

Some minor points:

1) In the molecular dynamics simulation, the authors removed the histidine-rich loop, and it is unclear if the created termini stayed open. If yes, it has a significant influence on the simulation, and it needs to be addressed.

Thanks for raising this point. The highly mobile 103-residues long histidine-rich loop (138-240) was removed from the MD initial model, with termini capped by acetyl and methylamide, in a similar setting comparable to other MD simulation investigations with long disorder loops truncated (Polovinkin et al., Nature, 2018; Billesbølle et al., Nature, 2020; Qu et al., Nature Chemical Biology, 2023). Our cellular assay showed that deletion of this loop decreased the ZnT1's efflux capacity, compared to WT ZnT1. This histidine-rich loop may help to increase local Zn^{2+} concentrations, allowing for

efficient entrance into the TMD translocation tunnel, as the intracellular free Zn^{2+} concentration is generally in the range of picomolar to nanomolar under physiological condition. To simulate the enrichment effect in MD, we placed several Zn^{2+} ions near the intracellular entry site of the inward-facing ZnT1 model. By doing so, we observed the Zn^{2+} entrance into the TMD substrate binding site of the inward-facing ZnT1, which allowed us to analyse the interplay between Zn^{2+} and key coordinating residues H43/D47/H251/D255 (Figure x9). While doing MD analysis on outward-facing ZnT1, we believed that the influence of this intracellular histidine-rich loop would be marginal, as it is distant from the TMD Zn^{2+} -binding site by the closure of the intracellular gate.

Figure x9. MD of Zn^{2+} entrance into the TMD of inward-facing ZnT1.

Notably, in a recent ZnT7 cryoEM structural study, Bui and co-workers observed a physical interaction between its 14-histidines loop and TMD substrate binding site in the inward-facing state (Bui et al., Nature Communications, 2023). While in our study, we didn't capture such engagement between its 7-histidines loop and TMD in various ZnT1 conformations. Such interaction may be a unique feature of ZnT7, as the number and distribution pattern of these histidine residues vary significantly between ZnT7 and ZnT1. However, we acknowledge that it is difficult to completely rule out any other potential effect of this histidine-rich loop on ZnT1, in the absence of direct evidence of physical interaction.

2) There is no discussion on the histidine-rich loop, which is one of the main differences between ZnT1 and other ZnTs.

Thanks for raising this point. We have conducted the cellular assay on ZnT1 mutant with the histidine-rich loop deleted ($\Delta 141-210$), which decreased ZnT1's efflux capacity, as shown in Figure x7. We now included the data and analysis in the text accordingly (Lines 198-209).

“Another feature in ZnT1 is the intracellular histidine-rich loop that connects TM4 and TM5 (Fig. 1A). Due to its intrinsic flexibility, modelling of this 7-histidines linker is not permitted (Fig. 2A). Interestingly, truncation of this histidine-rich loop ($\Delta 141-210$) also exhibited a small but significant effect on ZnT1-mediated Zn^{2+} transport (Fig. 2B). So far, such histidine-rich region is

limited to ZnT1 and ZnT7. Notably, Bui and co-workers observed a physical interaction between its 14-histidines loop and TMD substrate binding site in the inward-facing ZnT7 structure (Bui et al, 2023). This interaction may be unique to ZnT7, as the number and distribution pattern of these histidine residues vary significantly between ZnT7 and ZnT1 (Appendix Fig. S3).”

3) Figure EV4 C and D, additional panels should be added with the map around these helices. Since the transition between Helix 3-10 and canonical helix are one of the main points, it is not a match to the map, and what is the result of rotating the helix on the 3-10 forms other residues and other interactions.

As suggested, the additional panels of model-map fitting, including comparison of TM2 segments in both outward- and inward-facing Zn^{2+} -bound ZnT1, are now added (Figure x10). We now reorganized the Figures x3, x5 and x10 into Figure EV2, to conform the journal requirement.

Figure x10. MD of Zn^{2+} entrance into the TMD of inward-facing ZnT1.

The rotation of TM2 segment (41-47 residues) from canonical helix to 3_{10} form indeed introduced changes in the local interaction network. For instance, in the canonical form, His43 mainly contacts Zn^{2+} to form a coordination network with the residues D47/H251/D255, and is close to S46 (Figure x11A). When transitioned to the 3_{10} form, His43 swings away from Zn^{2+} -binding site and approaches to Glu102 on the neighboring protomer (Figure x11B). Cumulatively, the concerted movement from canonical helix to 3_{10} helix slightly expands the extracellular vestibule (Figure x11C), which may promote Zn^{2+} release.

Figure x11. Local environment changes between TM2 canonical helix to 3₁₀ helix.

Referee #2:

This paper represents a significant advancement in the zinc transporter research field by elucidating multiple states of zinc transporter 1 (ZnT1) through cryo-electron microscopy, capturing snapshots with zinc ions. Furthermore, molecular dynamics (MD) simulations analysis demonstrated that the protonation on His residues influences their interaction with the Zn ion. However, the paper has some weak points.

We thank the Reviewer for the insightful comments and strong support.

The mechanism by which proton-driven significant structural changes are elicited awaits further analysis. While mutational functional analysis *in vivo* was performed, definitive information on the molecular mechanisms was not obtained. Although lipid-related function is discussed in the later part of the paper, the assumption may want to be elucidated by measuring Zn transporter activity *in vitro*.

Thanks for the constructive suggestions to improve our manuscript. We agree with the Reviewer that our current mechanistic analysis on proton-driven structural changes is inadequate, which is critical for a comprehensive understanding of the H⁺/Zn²⁺ exchange. We suspected that His43 may be a critical proton sensor as it exhibits in different orientations in various conditions. Mutation of His43 did affect the transport activity. We admit that it is difficult to separate Zn²⁺ coordination ability from H⁺-sensing activity for His43. Moreover, how H⁺ facilitates ZnT1 transition from inward-facing to outward-facing remains unclear. However, due to limited revision time and competitiveness in the field, it is not feasible for us to obtain such definitive information.

The mechanism of ZnT1-mediated Zn²⁺ transport is more complicated than that of other ZnTs, as remarked by Reviewer #1. Indeed, ZnT1 has been characterized as an H⁺/Zn²⁺ exchanger (Shusterman et al., *Metallomics*, 2014), and studies showed that ZnT1 can also localize in cytoplasmic compartments including mitochondria and endosome (Sun et al., *American Journal of Physiology*, 2015; Yang et al., *Hepatology*, 2023), indicating similar scenarios for H⁺/Zn²⁺ exchange as other cytoplasmic ZnTs. In addition, recent biochemical analysis showed a Ca²⁺-dependence rather than a pH-dependence for ZnT1-mediated Zn²⁺ efflux (Gottesman et al., *Cell Calcium*, 2022; Sun et al., *Science Advance*, 2024). To understand the potential mechanism of Ca²⁺/Zn²⁺ exchange, we also determined a 3.52-Å ZnT1 structure with Ca²⁺ supplemented. Unfortunately, we could not identify any additional density corresponding to Ca²⁺ ion in the map. It is unclear if Ca²⁺ can bind to ZnT1 in the same way as its substrate Zn²⁺ does. Therefore, to minimize misunderstanding, we now change our manuscript title from “*Insights into the human zinc transporter ZnT1 mediated H⁺/Zn²⁺ exchange*” to “*Insights into the human zinc transporter ZnT1 mediated Zn²⁺ efflux*”.

Investigation of the potential lipid-related function in ZnTs is of interest. We had reconstructed ZnT3 WT and mutants in liposomes made of total brain lipid extract to mimic the natural synaptic vesicle environment, and used H⁺ fluorescent indicator HPTS or Zn²⁺ indicator membrane-impermeable FluoZin-3 to measure efflux activity. Unfortunately, the proteoliposomes exhibited undesirable high H⁺ leakage which hindered our further analysis. Future investigations may provide more insight into the tuning effect of lipids on Zn²⁺ transport.

While MD simulations focus on Zn binding sites, if structural changes in the transmembrane region were observed, it could further support the proposed model. At least, more elaboration on MD simulations about alpha helices movements would be beneficial.

Thanks for this constructive suggestion. We indeed observed the structural changes of canonical helix to 3₁₀ helix at TM2 segment during MD simulations as long as His43 was protonated. We now provided supplementary movies along with the revision (Movies EV1-EV6), and included analysis of potential effect on helical movement in the text as follows:

“In the outward-facing state of ZnT1, Zn²⁺ was stably coordinated by the non-protonated tetrahedral network His43/Asp47/His251/Asp255 throughout 1 μs duration (Fig. EV5A and Movie EV1), and the extracellular half of TM2 (residues 41-45) remained in the canonical helix shape. His43 sidechain was close to Ser46 and Zn²⁺ in the downward orientation. When His43 was protonated, His43 sidechain became mobile and distant from Zn²⁺, and TM2 segment shifted to a 3₁₀ helical shape (Movie EV2), resembling the substate captured by our cryo-EM analysis. We speculated that the partial positive charge conferred by protonation causes the weakly confined His43 to become repulsive against Zn²⁺ and significantly volatile, which introduces perturbation in the surrounding chemical environment and

transforms into concerted movement of TM2 segment..... Protonation on both His43 and His251 did not alter the positioning of Zn²⁺ (Movie EV3), suggesting additional protonation is required for zinc release from translocation passage. Indeed, when Asp47 and Asp255 were further protonated, the zinc coordination network was disrupted and Zn²⁺ was released (Fig. 6A, Movies EV4, EV5 and EV6)."

Addressing these points would improve the paper, but I encourage an early publication, given the impact of reporting multiple structures of the same protein. It might be appropriate to put unresolved issues in future studies.

We thank the Reviewer for the strong support of our work.

Minor Comments

Lines 49 and 58: use the same abbreviation: SLC30/ZnT or ZnT/SLC30.

Done as suggested. It is now written as SLC30/ZnT.

In the MD simulations, please provide an additional explanation regarding alpha helices changes.

Done as suggested.

"We speculated that the partial positive charge conferred by protonation causes the weakly confined His43 to become repulsive against Zn²⁺ and significantly volatile, which introduces perturbation in the surrounding chemical environment and transforms into concerted movement of TM2 segment."

In material and method,

Please provide the necessary information for constructing the proteins, including linkers and tags, which is essential for ensuring the reproducibility of the experiments.

Done as suggested.

Additionally, compiling a plasmid list would be helpful.

Done as suggested. The plasmid list file is now included in Source Data.

HEPES should be HEPES-Na or HEPES-K

Done as suggested.

Please provide the name of the maker of the DYKDDDDK(FLAG) affinity resin.

Done as suggested.

Lines 493 and 519: ZnSO₄: 4 should be subscript.

Done as suggested.

Figs. 1B and EV4: please indicate the range of electrostatic potential.

Done as suggested. To meet the journal requirements in terms of minimum/maximum number of figures, we have remade these figures. Fig. 1B is now changed to Fig. 1D; Fig. EV4 is now changed to Fig. EV2.

Fig. EV4C and B bottom right, please include an EM map.

Done as suggested. Fig. EV4 is now changed to Fig. EV2.

Figs. EV5 and EV6: please include sequence IDs for easy access.

Done as suggested. Uniprot ID for each sequence was included in the caption. Figs. EV5 and EV6 are now changed to Appendix Fig. S2 and S3.

Fig. EV7: for clarity and comparison, please specify the method of structure acquisition and the solubilization state (or embedded in nanodiscs). Additionally, please include RMSD and the regions that were superimposed.

Done as suggested. The update is included in Fig. EV3 and caption.

Fig. EV8: since there is no description of the Western blotting data, please include it.

Done as suggested. The description is now included in the caption of Fig. EV4A.

Fig. EV13: if possible, provide the accumulation data similar to EV8.

Done as suggested. The accumulation data is now included in Fig. EV4B.

Dear Qianhui,

Thank you once more for the submission of your revised manuscript to EMBO Reports and for your patience while it was reviewed.

I had already informed you about the remaining concerns from referee #1 that I kindly asked you to address. I have now also finished my checks from the editorial side and list below a few points that need to be addressed:

- Please do not include the figures in the manuscript file but only upload them as individual figure files.
 - Please provide up to 5 keywords.
 - Regarding the Author Contributions, we now use CRediT to specify the contributions of each author in the journal submission system. Therefore, please remove the Author Contributions from the manuscript file and make sure that the author contributions in our online manuscript tracking system are correct and up-to-date. The information you specified in the system will be automatically retrieved and typeset into the article. You can enter additional information in the free text box provided, if you wish.
 - All information on funding must be entered in our online manuscript tracking system. In this respect we note that the start-up funds from Shanghai Stomatological Hospital and School of Stomatology, Fudan University are missing.
 - Please add a callout for Fig. 3D in the text where appropriate.
 - In line 749 you cite Table S1. If I am not mistaken, this refers to Table EV1. Please update the callout.
 - Appendix: please provide page numbers in the Table of Content.
 - The individual cryo-EM densities in Appendix Figure S1 are rather small and since the resolution of the PDF is not that high and does not allow to zoom in that much, the details of the density maps are difficult to appreciate. Maybe you could split the figure into 2 or more to increase the size of the models?
 - Appendix Figure S5A: you provide the number of cells analysed, but please also specify whether the cells are from one experiment or from several replicates.
 - The Western blots in Figure EV4A and B appear to have low resolution. It appears as if an image with the blots and the lane labels had been copied and pasted into the figure file. Please improve the resolution and quality of the blots in this panel. The same is true for the graph in EV4B.
 - Please provide legends for each movie as a README.txt file. Then zip each legend together with its movie and upload the individual zipped files.
 - Author Checklist: please complete field D77. The question refers to whether you have used external Laboratory protocols, i.e., published step-by-step protocols.
 - Data availability section: Please insert URLs that resolve directly to the datasets at PDB and EMD.
 - The manuscript sections should be in the following order: Title page - Abstract & Keywords - Introduction - Results - Discussion - Methods - Data Availability - Acknowledgments - Disclosure Statement & Competing Interests - References - Figure Legends - (Main Tables with legends) - Expanded View Figure Legends.
 - Please correct the EV figure legends in the manuscript file: Figure EV1 instead of Expanded View Figure 1, etc.
 - Data availability section: We need specific URLs for 8XM6, 8Z4Z, 8XMA, 8XMF, 8XMJ, 8XN1, EMD-38465, EMD-39772, EMD-38469, EMD-38475, EMD-38479, EMD-38474, 809 and EMD-38494 datasets. i.e., the URLs should directly link to the datasets, not just the database itself.
 - Our production/data editors have asked you to clarify several points in the figure legends (see below). Please incorporate these changes in the manuscript and return the revised file with tracked changes with your final manuscript submission.
- A) Statistical test information. Only p-values that are actually shown in the figure panel(s) should (and must) be defined in the legends, all others should be removed from (or added to) the legend. Moreover, we ask for the specification of exact p-values:
- Please note that the exact p values are not provided in the legends of figures 2b; 6b.
 - Please note that in figures 2b; 6b; there is a mismatch between the annotated p values in the figure legend and the annotated

p values in the figure file that should be corrected.

- You cite three preprints in your manuscript and correctly labeled these with [PREPRINT] in the reference list. Please label also their in-text citation with 'preprint', e.g. (preprint: Rege et al, 2022).

- Please also add the preprint server (bioRxiv) to the citation in the citation list. E.g. Rege J, Nanba K, Bandulik S, Kosmann C, Blinder AR, Vats P, Kumar-Sinha C, Lerario AM, Else T, Yamazaki Y, et al (2022) Zinc transporter somatic gene mutations cause primary aldosteronism. bioRxiv doi:10.1101/2022.07.25.501443 [PREPRINT]

- Please do not forget to update your recent publication in Cell Research, which is still listed as preprint in your manuscript (Wang et al 2023, now PMID 38163846).

- Quick note: I spotted that you overlooked to label "DN" in the source data for ZnT3 WB, Figure EV4B.

- Since July this year we require that all Materials and Methods need to be described in the main text using our 'Structured Methods' format, which is required for all research articles. According to this format, the Methods section includes a Reagents and Tools Table (listing key reagents, experimental models, software and relevant equipment and including their sources and relevant identifiers) followed by a Methods and Protocols section describing the methods using a step-by-step protocol format. The aim is to facilitate adoption of the methodologies across labs. More information on how to adhere to this format as well as a downloadable template (.docx) for the Reagents and Tools Table can be found in our author guidelines: <https://www.embopress.org/page/journal/14693178/authorguide#structuredmethods>.

I am aware that your study was submitted prior to this deadline but also feel that the structured methods are helpful for other authors and that the Reagents and Tools table is not too much work to create.

- As a standard procedure, we edit the title and abstract of manuscripts to make them more accessible to a general readership. Please find the suggested versions below my signature.

- On a different note, I would like to alert you that EMBO Press offers a new format for a video-synopsis of work published with us, which essentially is a short, author-generated film explaining the core findings in hand drawings, and, as we believe, can be very useful to increase visibility of the work. This has proven to offer a nice opportunity for exposure i.p. for the first author(s) of the study. Please see the following link for representative examples and their integration into the article web page:

<https://www.embopress.org/doi/full/10.15252/emj.2019103932>

With kind regards,

Martina

Referee #1:

The author did an extensive revision of the paper. While they dealt with most comments and did a good job with the transport assay, they still have major issues with their map analysis. In my previous review I pointed to several structural issues which were only examples. The authors only fixed some of them but did not try to use the new correction to all the other structures, while in many cases, the maps are highly similar in these places.

One of the main point which is still not match their claims and is one of the key points in their paper is the movement between helix10 to canonical one. They also claim they re-sampled particles and that they have two subpopulations up and down in protomers A and B.

Unfortunately, I can't upload a screen shot to present the fit to the map, but I will give again few examples.

1: In 8XM6 above Trp74 there is a clear lipid, it should be added to the structure. Loop 392-299 does not fit well on the map. It can be seen better in 8XMA down up protomer A. It is not clear why this nice Met397 position is not used in all structures when the map is similar in this location. It means they need to fix this in most structures. Some rotamers are clearly inverted such Ile96 protomer A. Residue 242A and B are located into the map of the continuation of the main chain, and it seems there is another turn in the helix.

8XMA: The paper's main point is the location of H43 and the helix of 43-35. It is clear there is no very good fit of this helix in protomer A. When overlapping the whole structure of Protomer B on A it is clear Protomer B fits better and that there also a residual map for H43. Rotamer Thr319 A is inverted, Some residues are missing their side chains, such as Arg30B and Glu364A. Clearly 389-398B does not fit the map while A do. When overlapping A on B there is a good fit of the overlapped on the map. It has to be fixed as well to all other structures. Loop 443-438 does not fit well on the map, and it is clear that it is traced to match Cys 441 to Zn and not to the map. Based on all their maps, which share similar densities (maybe except 8XMF), it is clear that this Cys does not participate in Zn binding. Some rotamers are wrong, such as Leu51B, Arg62A Trp273A-missing side chains 111, 112A. In general, it is not clear why several side chains are missing and others not for the same quality of map coverage.

Other maps are low resolution, so there are fewer specific issues.

Overall, the maps are very good. The structures should be fitted better so that when deposited to the PDB, they describe the map correctly.

Once the structures and the corresponding discussion on Helix10 are fixed the paper should be accepted in my opinion.

Structural insights into human zinc transporter ZnT1 mediated Zn²⁺ efflux

Abstract

Zinc transporter 1 (ZnT1), the principal carrier of cytosolic zinc to the extracellular milieu, is important for cellular zinc homeostasis and resistance to zinc toxicity. Despite recent advancements in the structural characterization of various zinc transporters, the mechanism by which ZnTs-mediated Zn²⁺ translocation is coupled with H⁺ or Ca²⁺ remains unclear. To visualize the transport dynamics, we determined the cryo-electron microscopy (cryo-EM) structures of human ZnT1 at different functional states. ZnT1 dimerizes via extensive interactions between the cytosolic (CTD), the transmembrane (TMD), and the unique cysteine-rich extracellular (ECD) domains. At pH 7.5, both protomers adopt an outward-facing (OF) conformation, with Zn²⁺ ions coordinated at the TMD binding site by distinct compositions. At pH 6.0, ZnT1 complexed with Zn²⁺ exhibits various conformations [OF/OF, OF/IF (inward-facing), and IF/IF]. These conformational snapshots, together with biochemical investigation and molecular dynamic simulations, shed light on the mechanism underlying proton-dependence of ZnT1 transport.

Referee #1:

The author did an extensive revision of the paper. While they dealt with most comments and did a good job with the transport assay, they still have major issues with their map analysis. In my previous review I pointed to several structural issues which were only examples. The authors only fixed some of them but did not try to use the new correction to all the other structures, while in many cases, the maps are highly similar in these places.

One of the main point which is still not match their claims and is one of the key points in their paper is the movement between helix10 to canonical one. They also claim they re-sampled particles and that they have two subpopulations up and down in protomers A and B.

We thank the Reviewer for the constructive comments.

Unfortunately, I can't upload a screen shot to present the fit to the map, but I will give again few examples.

1: In 8XM6 above Trp74 there is a clear lipid, it should be added to the structure.

Done as suggested. Given the branched X-shape of these extra densities, we assigned them the detergent LMNG molecules which fit nicely in the density (Figure x1).

Figure x1. LMNG molecule fits in the density.

Loop 392-299 does not fit well on the map. It can be seen better in 8XMA down up protomer A. It is not clear why this nice Met397 position is not used in all structures when the map is similar in this location. It means they need to fix this in most structures.

Fixed as suggested.

Some rotamers are clearly inverted such Ile96 protomer A.

Fixed as suggested. Almost all rotamer outliers are now fixed in all structures.

Residue 242A and B are located into the map of the continuation of the main chain, and it seems there is another turn in the helix.

Done as suggested. A segment (238DRAG241) is now added.

8XMA: The paper's main point is the location of H43 and the helix of 43-35. It is clear there is no very good fit of this helix in protomer A. When overlapping the whole structure of Protomer B on A it is clear Protomer B fits better and that there also a residual map for H43.

We thank the Reviewer for this critical and valuable comment. We have now rebuilt this region for a slight improvement (Figure x2). Although we obtained two substate maps, as described in last revision, there could still exist some mixed particles or a conformational continuum in this short TM2 segment (residues 35-43), which may be a reflection of the residual density for H43. However, the stronger density for H43 sidechain pointing downward to the Zn^{2+} in Protomer A argues for a tetrahedral coordination network (H43/D47/H251/D255). When Zn^{2+} is coordinated in a tetrahedral network, TM2 of inward-facing ZnT1 assumes the canonical α -helix shape. It is possible that during the transition from inward to outward-facing state, TM2 could preserve the canonical α -helix shape for a certain period when Zn^{2+} is bound by H43/D47/H251/D255. Being that said, we agree with the Reviewer that Protomer A could fit better as a canonical α -helix to support our hypothesis.

We now add the following sentence in the text (Lines 247-249):

“It is worthy to mention that, this short segment is only constructed nearly to the ideal canonical α -helix shape, most likely due to residual conformational heterogeneity.”

Figure x2. TM2 conformational difference in Zn^{2+} -bound outward-facing (A,B) and inward-facing ZnT1 (D) models. Overlays (C, E) are indicated correspondingly.

Rotamer Thr319 A is inverted,

Fixed as suggested. Almost all rotamer outliers are now fixed in all other structures.

Some residues are missing their side chains, such as Arg30B and Glu364A.

Done as suggested.

Clearly 389-398B does not fit the map while A do.

Fixed as suggested.

When overlapping A on B there is a good fit of the overlapped on the map. It has to be fixed as well to all other structures.

Done as suggested.

Loop 443-438 does not fit well on the map, and it is clear that it is traced to match Cys 441 to Zn and not to the map. Based on all their maps, which share similar densities (maybe except 8XMF), it is clear that this Cys does not participate in Zn binding.

Fixed as suggested.

Some rotamers are wrong, such as Leu51B, Arg62A Trp273A-missing side chains 111, 112A.

Fixed as suggested.

Rotamer flip: Thr402,

Fixed as suggested. And side chains are now added to several residues (Gln111, Gln112, Leu114, Arg246, Arg361, Lys401, Gln350, Ile351, Val347, Arg354, Lys358, Glu359, Leu360, Glu366, Ile105, Glu338)

In general, it is not clear why several side chains are missing and others not for the same quality of map coverage.

Fixed as suggested. Side chains are added whenever density permits.

Other maps are low resolution, so there are fewer specific issues.

Overall, the maps are very good. The structures should be fitted better so that when deposited to the PDB, they describe the map correctly.

Agreed. Done as suggested.

Once the structures and the corresponding discussion on Helix10 are fixed the paper should be accepted in my opinion.

Again, we are grateful to the Reviewer for the improvement of manuscript and support of our work. We acknowledge that the TM2 in Promoter A is not constructed

as an ideal canonical α -helix when Zn^{2+} is chelated in the tetrahedral network, possibly due to the residual conformational variability. However, given that TM2 can adopt the canonical α -helix shape in the Zn^{2+} -bound inward-facing ZnT1, or the typical 3_{10} helix shape in the outward-facing Zn^{2+} -bound D47/H251/D255 coordination of Protomer B or the outward-facing Zn^{2+} -free Apo structure, it is conceivable that TM2 would experience a transition between canonical α -helix and the 3_{10} helix state during the TMD rocking-bundle conformational rearrangement from inward- to outward. It would be interesting to obtain some discrete intermediate states out of the conformational continuum with additional high-quality data or more sophisticated data processing strategies in future.

Manuscript number: EMBOR-2024-58919V3

Title: Structural insights into the human zinc transporter ZnT1 mediated Zn²⁺ efflux

Author(s): Qianhui Qu, Yonghui Long, Zhini Zhu, Zixuan Zhou, Chuanhui Yang, Yulin Chao, Yuwei Wang, Qingtong Zhou, and Ming-Wei Wang

Dear Dr. Qu

Thank you for your patience while we have reviewed your revised manuscript. As you will see from the report below, referee 1 is positive about its publication in EMBO reports.

I am now writing with an 'accept in principle' decision, which means that I will be happy to accept your manuscript for publication once the referee concerns and a few minor corrections from the editorial side have been addressed, as follows.

1) Please address the remaining concerns from referee1 by carefully phrasing your conclusions and/or by removing the data in question and please also provide a point-by-point response.

2) In Figure 2b one of the p-values seems missing. There are 6 p-values annotated in the figure panel but only 5 defined in the legend. One of the ** values seems missing.

3) In Figure 6b you state: "Significance was analyzed by one-way ANOVA with Turkey post-hoc test. *p<0.05, **p<0.01, ***p<0.001, ****p<0.0001, ns=non-significant. p=0.0004, 0.0004, 0.5882, 0.3716, 1.9×10⁻⁵."

However, the figure panel only shows ns, *** and ****. Therefore, the legend text should be modified to annotate only the p-values that are shown in the figure:

"Significance was analyzed by one-way ANOVA with Turkey post-hoc test. ***p<0.001, ****p<0.0001, ns=non-significant. p=0.0004, 0.0004, 0.5882, 0.3716, 1.9×10⁻⁵."

Once you have made these minor revisions, please use the following link to submit your corrected manuscript:

Link Not Available

If all remaining corrections have been attended to, you will then receive an official decision letter from the journal accepting your manuscript for publication in the next available issue of EMBO reports. This letter will also include details of the further steps you need to take for the prompt inclusion of your manuscript in our next available issue.

Thank you for your contribution to EMBO reports.

Yours sincerely,

Referee #1:

While the authors did their best to deal with the structural fitting to the map, some points stayed to be addressed. apo 8XMG 294-305 does not represent the map at all and should stay deleted as the previous submission. same as for residue 441, which clearly does not point to the zinc as in the 8z4z structure in high res. This is a recurrent problem that the 436-447 has different conformation in different structures while the high-res maps are all similar, and the low-res map will fit well to that one conformation generated in the high-res map.

The key point of Helix 10 should be removed from the story. I understand the appeal of this to their mode of operation, but it has not been supported by the maps. In 8xmg, the map still fits better to the normal helix (just overlapped chain A on B). Also, in 8XMA, the map fits the canonical helix, and as an indication, the authors built a normal helix in the 8z4z that its map overlaps perfectly with the 8XMA map. It seems that the extra density near the Zn can be ligand or water but is less likely from His residue, which is dragged to such a location and alters the backbone of its helix, which is clearly out of the map.

Referee #1:

While the authors did their best to deal with the structural fitting to the map, some points stayed to be addressed.

We are grateful to the Reviewer's critical and instructive comments.

apo 8XMG 294-305 does not represent the map at all and should stay deleted as the previous submission. same as for residue 441, which clearly does not point to the zinc as in the 8z4z structure in high res. This is a recurrent problem that the 436-447 has different conformation in different structures while the high-res maps are all similar, and the low-res map will fit well to that one conformation generated in the high-res map.

As suggested by the Reviewer, we now removed the 294-305 region and corrected the pose of residue 441 of apo 8XM6 model and other low-res maps according to the conformation in high-res maps for deposition. We apologize for the unintended errors.

The key point of Helix 10 should be removed from the story. I understand the appeal of this to their mode of operation, but it has not been supported by the maps. In 8xmg, the map still fits better to the normal helix (just overlapped chain A on B). Also, in 8XMA, the map fits the canonical helix, and as an indication, the authors built a normal helix in the 8z4z that its map overlaps perfectly with the 8XMA map. It seems that the extra density near the Zn can be ligand or water but is less likely from His residue, which is dragged to such a location and alters the backbone of its helix, which is clearly out of the map.

We appreciate the Reviewer's critical comments on the point of helix \$3_{10}\$ /normal helix transition of a short TM2 segment (residues 41-46). Despite two new maps after extensive classification on a 2.65-Å \$Zn^{2+}\$ -bound outward-facing ZnT1 map in previous submission, we agreed with the Reviewer that the TM2 segment did not fit perfectly into the map as a canonical helix. Therefore, we modelled the two protomers similarly in the 2.65-Å map as suggested by the Reviewer. We now removed all the descriptions or discussions on Helix 10 from the text, and made new figures accordingly.

Dr. Qianhui Qu
Institutes of Biomedical Sciences
No.131 Dongan Road, Xuhui District
Shanghai 200032
China

Dear Dr. Qu,

Thank you for implementing the final minor edits. I am very pleased to accept your manuscript for publication in the next available issue of EMBO reports. Thank you for your contribution to our journal.

Yours sincerely,
